



# A methodology to estimate flow duration curves at partially ungauged basins

Elena Ridolfi[1], Hemendra Kumar[2], and András Bárdossy[3]

[1]Department of Civil and Environmental Engineering, University of Perugia, Perugia, Italy
[2]Department of Biosystems Engineering Auburn University, Alabama, USA
[3]Institute of Hydraulic Engineering, University of Stuttgart, Stuttgart, Germany

**Correspondence:** András Bárdossy (andras.bardossy@iws.uni-stuttgart.de)

**Abstract.** The Flow Duration Curve (FDC) set up at a specific site has a key role to the knowledge of the streamflow characteristic at that site. The FDC gives information on the water regime providing important information to optimally manage the water resources of the river. Spite of its importance, because of the lack of streamflow gauging stations, the FDC construction can be a not straightforward task. In ungauged or partially gauged catchments, FDCs are usually built using regionalization

methods among the others. In this paper we show that the FDC is not a characteristic of the basin only, but of both the basin and the weather. Different weather conditions lead to different FDC for the same catchment. The differences can often be significant. Similarly, the FDC built at a site for a specific period of time cannot be used to retrieve the FDC at a different site for the same time window.

     In this paper we propose a new methodology to estimate FDCs at partially gauged basins (i.e., target sites) using discharge

and precipitation data gauged at another catchment (i.e., reference catchment). The main idea is that it is possible to retrieve the FDC of a target period of time using the data gauged during a given reference time period for which data are available at both target and reference sites.

     To test the methodology, several reference and target time periods are analysed and results are shown for two different case study areas. The comparison between estimated and actually observed FDCs show the reasonability of the approach especially

for intermediate percentiles.

## 1 Introduction

A duration curve is a function that associates to a specific variable its exceedance time. Specifically, in hydrology a Flow Duration Curve (FDC) is a function describing the flow variability at a specific site during a period of interest. It represents the streamflow values, gauged at a site, against their relative exceedance time. An empirical long-term FDC is the complement of

the empirical cumulative distribution function of streamflow values at a given time resolution based on the complete streamflow record available for the basin of interest (Castellarin et al., 2007). FDCs are built as explained in the followings:

- rank the streamflow values in descending order;

- plot the sorted values against their corresponding frequency of exceedance.





The duration $d_i$ of the $ith$ sorted observation is its exceedance probability $P_i$. If $P_i$ is estimated using a Weibull plotting position, the duration $d_i$ for any $q_i$ (with $i = 1, ..., N$) is

$$d_i = P(Q < q_i) = P_i = \frac{i}{N+1}, \tag{1}$$

where $N$ is the length of the streamflow series and $q_i$ is the $ith$ sorted streamflow value.

The FDC provides historical information on the water regime: on the severity of the droughts and on the magnitude of high flows. Several time resolutions of streamflow data can be used to build the FDC: annual, monthly or daily. However, the finer is the resolution, the higher is the information provided by the FDC about the hydrological characteristics of the river (Smakhtin, 2001). FDCs may be built either on the basis of the whole available record period (Vogel, 1994); or on the basis of all similar

months (Smakhtin et al., 1997); or on the basis of a specific month.

FDCs have a key role in hydrology for water resources analysis as testified by the variety of applications reported by Vogel (1994). For instance, FDCs are at the base of hydropower plants design as they are used to determine the hydropower energy potential, especially for run-of-river plants (Hänggi and Weingartner, 2012; Blöschl et al., 2013).

Through FDCs it is possible to evaluate the available flows for intake purposes. Indeed, the FDC provides information about

the reliability of the water resource for water abstraction activities (Dingman, 1981). As FDCs are a key signature of runoff variability, they can be used to assess the impact of changes in a catchment. To this end, Vogel et al. (2007) introduced the indicators of the eco-deficit and eco-surplus estimated on the basis of the FDC. Moreover, FDCs can be used to define and investigate low flows (Smakhtin, 2001).

The knowledge of the streamflow characteristics is also relevant for stream water quality studies, for instance, to regulate

the proper threshold for chimical concentration and load (Bonta and Cleland, 2003). FDCs have a further application in model calibration. The methodology is based on the replication of the flow frequency distribution rather than of the simulation of the hydrograph (Yu and Yang, 2000; Westerberg et al., 2011).

Other applications are related to irrigation planning (Chow, 1964); schedule optimal flow release from reservoirs (Alaouze, 1991); basins afforestation (Scott et al., 2000); investigation of the effects on flows regime due to catchments vegetation change

(Brown et al., 2005).

Spite of FDCs importance, FDCs are affected by the lack of data in ungauged and poorly gauged basins. Many authors dealt with the issue of FDC prediction at ungauged or partially gauged locations through regional regression (e.g., Fennessey and Vogel, 1990; Mohamoud, 2008; Rianna et al., 2011, 2013; Castellarin et al., 2013; Pugliese et al., 2016) and geostatistical interpolation (e.g., Pugliese et al., 2014) techniques. Ganora et al. (2009) developed a methodology to estimate FDCs at

ungauged sites based on distance measures that can be related to the catchment and the climatic characteristics. Moreover, the determination of streamflow characteristics at ungauged sites was one of the main objective of the IAHS's initiative in the Predictions in Ungauged Basins (PUB) decade framework (Sivapalan et al., 2003).

In this work we present a methodology to predict FDCs in poorly gauged and ungauged basins using a transformation based on the corresponding weather conditions. Analysis show that the FDCs built at a given location for different periods of time





cannot be regarded as the same distribution. This implies that it is not possible to use the same FDC when dealing with different time windows. Moreover, it is not possible to determine a unique distribution and therefore a unique set of parameters.

This also implies that the FDC is not a property of a specific basin, because different time windows (such as decades) relate

to different FDCs.

The same results from the analysis of FDCs built in two different catchments. It is not possible to use the distribution of a FDC built at a location to determine the distribution of the FDC at another location based on catchment properties only.

These issues have a key role especially when dealing with partially and completely ungauged basins. Indeed, there still remains the question of how the FDC can be determined in case of missing streamflow data. Here we want to define a method-

ology to build FDCs facing the aforementioned issues. The main idea is that it is possible to build a FDC for a given target period using a transformation driven by the weather. The paper is organized as follows. First, the case studies are presented and catchments are grouped into energy- and water-limited ones. Then, the Kolmogorov-Smirnov test is carried out on pairs of FDCs to assess whether these curves can be regarded as the same distribution. Second, the methodology is presented and applied to a set of catchments located in Germany and in U.S. Finally, results are shown and discussed.

## 15  2   Case study area

The methodology has been applied to several catchments located in two different areas. Ten basins are located in the upper Neckar River basin (Germany), while ten basins are located on the East U.S.A. coast. In the followings the two case study areas are presented. Since the procedure is based on the climatological characteristics of basins, catchments will be divided in water and energy limited ones.

### 20  2.1   Upper Neckar catchments (Germany)

This study uses data from ten sub-catchments belonging to the Upper Neckar River basin, south-west Germany. The Neckar is a tributary of the Rhine, it springs at an altitude of 706 m a.s.l. and it is 367 km long, Figure 1. The Upper Neckar catchment lies in between the Black Forest and Schwäbische Alb in the Baden-Württemberg region. The Upper Neckar basin has an area of $4000 \, \text{km}^2$, its elevation ranges from about 240 m a.s.l. to around 1010 m a.s.l., with a mean elevation of 548 m a.s.l. (Singh et al.,

2012). The sub-catchments are characterized by a drainage area ranging from around $120 \, \text{km}^2$ to about $4000 \, \text{km}^2$. The region is characterized by warm summers and mild winters (Samaniego, 2003). In the Upper Neckar catchments, the main geological formations originated in the Triassic and Jurassic periods. The main formations are composed of altered keuper, claystone-jura, claystone-keuper, limestone-jura, loess, sandstone and shelly limestone (Muschelkalk), Samaniego (2003). The effect of soil type can strongly modify the impact of climate on the water balance. For instance, karstic and non-karstic catchments are

characterized by very different water balances, since an underground karstic catchment is very different from its overground catchment. The presence of karstic regions makes difficult the transfer of information from precipitation to discharge data in the same basin and from a karstic basin to a not karstic one. Approximately 35% of the basin has karstic formations (Samaniego et al., 2010).





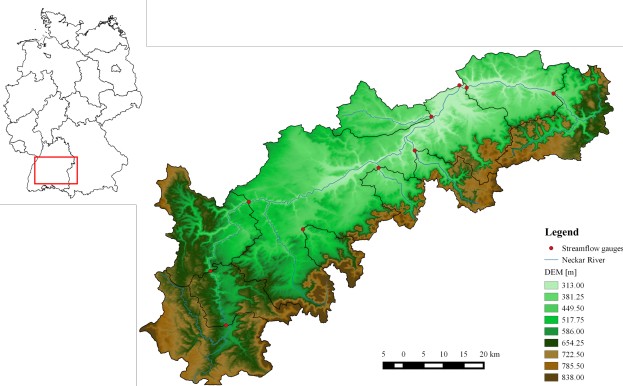

**Figure 1.** Streamflow gauges (green circles) used to test the methodology in the corresponding catchments located on the Upper Neckar River, Germany.

The mean daily discharge, precipitation, and evapotranspiration, the minimum and maximum daily temperature are available for each sub-catchment for the period 1961-1990. Basins characteristics are presented in Table 1; for more details on this study area, please refer to Samaniego (2003) and Bárdossy et al. (2005).

**Table 1.** Study area: upper Neckar catchments in south-west Germany.

| Catchment | Area km$^2$ | Drainage area km$^2$ | Elevation m | Slope degree | Annual discharge mm | Annual precipitation mm |
|---|---|---|---|---|---|---|
| Rottweil | 456 | 456 | 555–1010 | 0–34.2 | 352.7 | 976 |
| Obendorf | 235 | 691 | 460–1004 | 0–44.2 | 360.5 | 953 |
| Horb | 427 | 1118 | 383–841 | 0–48.9 | 417.5 | 1158 |
| Rangendingen, Starzel | 118 | 118 | 421–954 | 0–36.9 | 347.4 | 905 |
| Wannweil, Echaz | 135 | 135 | 309–862 | 0–45.9 | 654.1 | 877 |
| Riederich, Erms | 170 | 170 | 317–865 | 0–49.4 | 556.5 | 956 |
| Oberensingen, Aich | 175 | 175 | 278–601 | 0–27.1 | 234.3 | 762 |
| Suessen, Fils | 340 | 340 | 360–860 | 0–49.3 | 547.2 | 1003 |
| Plochingen, Fils | 352 | 692 | 252–785 | 0–39.7 | 446.6 | 936 |
| Plochingen, Neckar | 473 | 3962 | 241–871 | 0–45.8 | 397.2 | 863 |

5 ## 2.2 U.S.A. catchments

The catchments on the East coast of the U.S.A. are located in three different States: Florida, Louisiana and Texas, Figure 2. These basins were selected because they are characterised by a mild climate and therefore, no snow events have been recorded, allowing us to neglect the snow melting effect. Daily streamflow discharge and precipitation values are available for





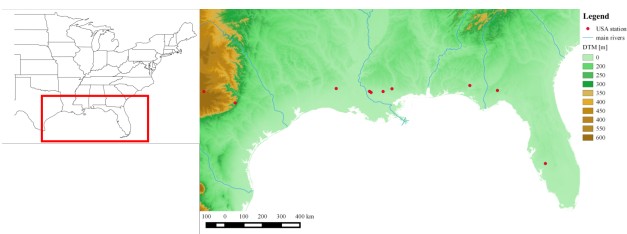

**Figure 2.** Streamflow gauges (red circles) used to test the methodology in the corresponding U.S.A. catchments.

each catchment for different time windows, Table 2. Daily streamflow discharge data were originally provided by the United States Geological Survey (USGS) gauges, while mean areal precipitation and climatic potential evaporation were supplied by the National Climate Data Center (NCDC) at daily resolution. The data set is a subset of the Model Parameter Estimation Experiment (MOPEX) dabase, used for hydrological model comparison studies (Duan et al., 2006) and for simultaneous calibration of hydrological models (Bárdossy et al., 2016).

**Table 2.** U.S.A. case study area: streamflow gauges and corresponding catchments characteristics.

| Station name | Drainage Area | Mean elevation | Mean slope | Annual discharge | Annual precipitation | Available record |
|---|---|---|---|---|---|---|
| | km$^2$ | m | − | mm | mm | − |
| Peace Rv At Arcadia, FL | 3540.53 | 32.3 | 0.3 | 257.4 | 1296.2 | 1948-2001 |
| Ochlockonee Rv near Havana, FL | 2952.6 | 75.6 | 1.8 | 322.6 | 1366.7 | 1948-2001 |
| Choctawhatchee Rv at Caryville, FL | 9062.41 | 92.2 | 3.2 | 540.8 | 1464.7 | 1948-1994 |
| Bogue Chitto Rv near Bush, LA | 3141.67 | 101.6 | 1.8 | 579.2 | 1637.1 | 1948-1999 |
| Tangipahoa Rv at Robert, LA | 1673.14 | 76.9 | 1.6 | 635.2 | 1682 | 1948-1999 |
| Comite Rv near Comite, LA | 735.56 | 59.6 | 1.1 | 595.9 | 1644.2 | 1948-1999 |
| Amite Rv near Denham Springs, LA | 3315.2 | 75.6 | 1.3 | 584.1 | 1647.9 | 1948-1999 |
| Calcasieu Rv near Oberlin, LA | 1950.27 | 62.2 | 1.1 | 502.9 | 1558.9 | 1948-1986 |
| Llano Rv near Junction, TX | 4807.04 | 670.9 | 3.4 | 34.8 | 645.8 | 1948-1988 |
| Blanco Rv at Wimberley, TX | 919.45 | 417.3 | 5.2 | 140.6 | 896.7 | 1948-2001 |

## 2.3 Energy and water limited catchments

Annual runoff variability is driven by the relative availability of water (i.e., precipitation) and energy (i.e., evaporation potential), Blöschl et al. (2013). As a consequence, the weather is the most important driver of annual variability. Much of the annual runoff variability can be explained observing the different availability of water and energy. For instance, if more water arrives to the catchment than energy can remove through evaporation, the annual runoff will be high. Moreover, in this case the relationship between runoff and precipitation will be more linear than when more energy is available to evaporate the



water. On the other hand, in a arid region, the aridity of the climate determines a high inter-annual runoff variability because of the non-linear relationship between runoff and precipitation. Therefore, differences in water and energy availability cause differences in annual runoff variability. However, additional factors such as differences in seasonality and precipitation must be considered (Jothityangkoonad and Sivapalan, 2009). The relative availability of water and energy can be described through the Budyko curve (Budyko, 1974). The curve plots the ratio between mean annual actual evaporation and mean annual precipitation as a function of the ratio between mean annual potential evaporation and mean annual precipitation (i.e., the aridity index). Therefore, it defines a similarity index (i.e., the aridity index) to express the availability of water and energy, and thus bolsters the classification of hydrological sceneries into various degree of aridity. The Budyko curve represents the effects of water and energy availability on annual runoff variability. Moreover, it provides indication about the synchrony of evaporation and precipitation. For instance, where precipitation and evaporation are in phase, runoff production reduces since the catchment infiltrates and stores water and vice versa. Many regions range from in phase to out of phase because of the strong seasonality of climate forcing. However, also the climatic timing can impact runoff variability as presented by Montanari et al. (2006). They shown that the difference in annual runoff between two years with equivalent annual precipitation was of 100% in a monsoonal area of Northern Australia because during the wet year the precipitation occurred during the wet season, i.e., when the potential evaporation was smaller.

In this framework, it is important to understand the behaviour of the catchments under analysis. To this end, we analysed the mean annual runoff coefficient, the annual precipitation and the annual evapotranspiration against the annual mean temperature. This analysis is essential to understand the causal processes leading to the long-term mean and variability of runoff as also described in McMahon et al. (2013). The mean annual runoff coefficient is defined as:

$$\mu_R = \frac{\bar{Q}_{yr}}{\bar{P}_{yr}}, \qquad (2)$$

where $\bar{Q}_{yr}$ is the annual discharge volume and $\bar{P}_{yr}$ is the annual precipitation volume. Results show that catchments have two different behaviours: precipitation, evapotranspiration and runoff have either a positive or a negative correlation with the air temperature. In the former case the evapotranspiration is limited by the available water which happens in water-limited catchments; in the latter the evapotranspiration is limited by the available energy which happens in energy-limited catchments. For instance, measurements at Peace River (LA) suggest that the catchment is water-limited, Figure 3 upper panel. While Ochlockonee River (FL), Amite River near Denham Springs (LA) and Bogue Chitto River (LA) are energy-limited, Figure 3. Results for Amite River are consistent with what found by Carrillo et al. (2011). Since it is not possible to infer discharge values of a water-limited catchment from the data set of an energy-limited one, analysis have been carried out on climatically homogeneous sets of basins.





**Figure 3.** Annual precipitation against mean annual temperature (left panel), annual evapotranspiration against mean annual temperature (middle panel) and annual runoff coefficient against mean annual temperature (right panel) for four different catchments: Peace River (FL), Ochlockonee River (FL), Amite River near Denham Springs (LA), Bogue Chitto River (LA). In each plot, the Pearson correlation coefficient $\rho$ is reported in box.



## 3   Methodology

The FDC can be interpreted as distribution function of discharge over a given time period. To determine if samples are drawn from the same distribution, the Kolmogorov-Smirnov test is carried out on each pair of samples. Then, in the followings, the methodology proposed in this paper to build a FDC at a location using precipitation and discharge data gauged at another location is described step by step. Since the methodology involves the use of the API index, it will be introduced in this section.

### 5   3.1   Application of the Kolmogorov-Smirnov test to FDCs

To determine if two FDCs built either at a location in different periods or at two different locations in the same period of time could be regarded as the same distribution, the Kolomogorov-Smirnov (KS) statistic is performed. The KS statistic on two samples is a non-parametric test for the null hypothesis that the two independent samples are drawn from the same continuous distribution. The decision to reject the null hypothesis is based on comparing the p-value with the significance level alpha. Here, alpha is set equal to 5%.

The KS statistic is applied on daily streamflow data sampled in several periods of record (e.g. 1 year, 10 years, 15 years). The test is carried out both on pairs of samples gauged at the same location in two different years (or in two different decades) and on couples sampled at two different sites. Since the streamflow data presents autocorrelation, the autocorrelation effects the KS test. Thus, we need to be more strict on the rejection of the null hypothesis. We can assume that the autocorrelation effect attenuates after three days. However, if the samples were 3 times smaller and for instance their length would equal 122 (i.e., 365 divided by 3), the null hypothesis would have been rejected anyway, leading to the same conclusion (i.e., the two samples cannot be regarded as the same distribution). This is due to the fact that, according to the two samples KS test, the length of the equivalent sample that could pass the test should be 22.

Results show that streamflow data gauged in different period of time (e.g. years or decades) at a specific location do not have the same distribution. The consequence is that it is not possible to use the parameters and the distribution derived from a FDC built for a specific time window to build the FDC of another time window. The same results comparing streamflow data gauged in a specific year or decade at two different sites. Since the two data sets cannot be regarded as the same distribution, it is not possible to derive the FDC at one location using the parameters of a FDC sampled at another location. Figure 4 shows how different can be FDCs built at the same location using streamflow data gauged during different time windows.

### 25   3.2   The Antecedent Precipitation Index

As resulted from the KS test, FDCs cannot be considered an invariant characteristic of a basin. The fact that FDCs are not invariant suggests that the weather is a driver of annual runoff variability. Indeed, the reason of the invariance must be the weather conditions as others (e.g. the catchment area, the land use) did not change. The most important descriptor of the weather characteristic is the rainfall, however, rainfall data series are characterized by a high amount of zeros which would affect the methodology presented here. Therefore, we need a weather descriptor which is not affected by ties. The methodology proposed in this paper involves the use of the Antecedent Precipitation Index (API). This index allows to take into account the





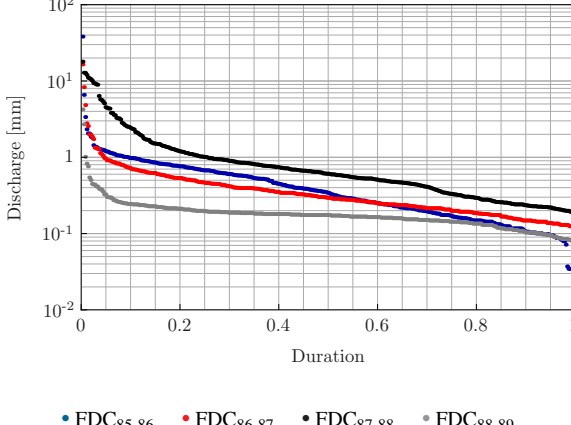

**Figure 4.** FDCs built for Tangipahoa River (FL) for four different years. Every year starts in October and ends the following September.

antecedent conditions, the duration of the rainfall events and gives an estimate of the portion of rainfall contributing to storm runoff (Linsley et al., 1949). It is a sequence of linear combination of rainfall events in the period preceding a specific storm (Kohler and Linsley, 1951). It represents the memory of a basin as it is related to the amount of water released by the soil to the river considering a specific time window. The API is used to investigate precipitation data in a similar way to discharge data. For a resolution of one day and a time window of 30 days, API at the $ith$ day is given by:

$$\text{API}_i = \sum_{j=0}^{29} \alpha^j P_{i-j}, \tag{3}$$

where $\alpha$ is a constant and ranges from 0 to 1 and $P_i$ is the daily precipitation occurred at the $ith$ day. Since a day-to-day value of the API is required, there is a considerable advantage in assuming $\alpha$ decreasing with the time as shown by Kohler and Linsley (1951). When $\alpha$ tends to zero, API keeps tracks of the precipitation occurred in the few previous days and it represents the short memory of the basin. When $\alpha$ tends to 1, API represents the long memory of the basin as it includes the effect of precipitation occurred many days before. To capture this behaviour, in this study $\alpha$ is chosen equal to 0.85.

### 3.3 How to determine the FDC at a partially gauged basin

Since it is found that two FDCs built for the same period of time at two different sites cannot be regarded as the same continuous distribution, we present here a methodology to overcome this issue and to estimate the FDC at an ungauged site. The FDC at one site is determined using climatological data gauged at another site. The FDC is expressed in millimeter, thus the area of the catchment is not an issue while retrieving the FDC from data of another catchment. The methodology is carried out under the assumption of large scale precipitation.

Let consider two catchments, A and B. We want to determine the Flow Duration Curve at catchment B for each time period from data available at catchment A. Therefore, A will be named reference catchment, while B will be called target catchment.



Specifically, the procedure involves daily Antecedent Precipitation Index (API) and daily streamflow values of both A and B. The API is considered as a support variable to obtain the FDC at the ungauged site from the FDC of the gauged one.

Let suppose that in a given number of years, catchment B is characterized by a lack of streamflow data, while data are available for A.

The procedure to predict streamflow values in catchment B is explained step by step in the followings.

1. Select a number of years for which discharge and precipitation values are available at daily resolution for both catchments (e.g. duration of 1 year, 10, 15, 20 years). These will be named reference years.

2. From the precipitation values, estimate the API. Sort API and discharge values of each basin in descending order and assign to each sorted value the corresponding rank $i$, with $i = 1, ..., N_r$ where $N_r$ is the length of each reference data series.

3. For both basins, from the rank values, determine the exceedance probability of each API value as $\frac{i}{N_r+1}$. In the same way, for both catchments, find the exceedance probability of each discharge value.

4. Select a period of time (e.g. 1 year, 10, 15, 20 years) in which streamflow values are missing at catchment B, while they are available at A. This period of time is then called target period and has length equal to $N_t$.

5. For basin A, sort the API data series of the target period in descending order. Assign to each sorted value the correspond-
ing rank and evaluate the associate exceedance probability.

6. In order to determine the streamflow associated to a given probability of exceedance $P_i$ (with $i = 1, ..., N_t$) occurred in the target period at site B, take the API value with probability equal to $P_i$ occurred at site A during the target year. This will be named $API_{Ati}$.

7. From the sorted API data series occurred during the reference period at site A, find the API value equal to $API_{At_i}$. This
API value has a probability of exceedance equal to $P_j$ and is then called $API_{Ar_j}$ (with $j = 1, ..., N_r$).

8. From the sorted streamflow data series gauged at site B during the reference period, find the discharge value $q_{Br_j}$ having exceedance probability equal to $P_j$ in the reference period. $q_{Br_j}$ is the value of discharge occurred during the target year at site B with a probability of exceedance equal to $P_i$.

If in the reference period at site A, it does not exist a value of $API$ equal to $API_{At_i}$, then, from the sorted API data series
occurred during the reference period at site A, take the ranks of the two most similar values to $API_{At_i}$, one should be bigger and the other smaller than $API_{At_i}$ (i.e., $API_{Ar_j}$ and $API_{Ar_{j+1}}$ having probability of exceedance $P_j$ and $P_{j+1}$, respectively). In the sorted streamflow data series gauged at site B during the reference period, look for the two discharge values of B having exceedance probability equal to $P_j$ and $P_{j+1}$, respectively. Evaluate the mean value of these discharge values to obtain the discharge occurred during the target year at site B having probability of exceedance equal to $P_i$.

The procedure is repeated for all exceeding probabilities $P_i$ with $i = 1, ..., N_t$. Therefore, the streamflow value at site B is





determined at each duration for the chosen target period. From the streamflow values obtained for the specific target period, the corresponding FDC is plotted. This will be named simulated FDC. The simulated streamflow values at each percentile (i.e., streamflow value associated to a specific duration) are compared with the observed values, that were actually available at site B.

Similarly, the procedure is repeated for every target period and for different target catchments. In this paper, the procedure is carried out using either API values or discharge values as support variable. For the German and U.S. catchments, results are shown using discharge and API as support variable, respectively. An example of the procedure used for the U.S. catchments is reported step by step in Appendix A.

## 4  Results

The procedure explained above has been tested on several target catchments varying both reference and target time periods. For the German catchments when a target time periods of 20 years is considered and the reference period equals 10 years, this case will be later named Test 1. Test 2 is built considering as both reference and target time periods 15 years, while for Test 3, the target time period equals 10 years and the reference one equals 20 years.

For the U.S. catchments, using a reference period of 20 years, we considered 10 years and 1 year as target periods. For reference periods equal to 15 and 10 years, we considered as target periods 15 and 10 years, respectively. The reference catchment is Blanco River (TX), while the target catchments are Tangipahoa (LA), Bogue (LA), Choctwhatchee (FL), Ochlocknonee (FL), Comite (LA), Amite (LA), Calcasieu (LA), they are all energy-limited catchments.

Results show a good agreement between observed and simulated FDCs, especially when a larger data set is used as both target and reference periods.

The FDCs simulated using 20 and 10 years as reference and target periods, respectively, show a good agreement with observed FDCs, as shown for Tangipahoa and Bogue catchments, Figure 5. It is interesting to note that observed and simulated FDCs are similar in shape and in values. The agreement is higher for intermediate durations, while in some cases it can be smaller for the droughts as at Bogue for target years 1988-1998 (Figure 5 lower panels) and for the floods as it can be observed at both Tangipahoa and Bogue sites, Figure 5.

The higher agreement in shape than in value is more evident when a small data set is considered as target period. For instance it results when the target period is 1 year, Figure 6. It is interesting to note that, even if for some target periods the simulated FDC does not perfectly match the observed one, the two FDCs have the same shape. For instance, this can be observed at Tangipahoa River for all but one (i.e., 1969-1970) target years, Figure 6.

In these cases, there is a shift between the two curves; the difference could be due to the different temperature values characterizing the reference and the target basins. This effects the evapotranspiration in the two basins and therefore, the streamflow values.

On the other hand, it is note worthy that when larger time periods are considered both as target and as reference, the simulated and observed FDCs almost perfectly match, Figure 7, for 15 years and Figure 8 for decades.




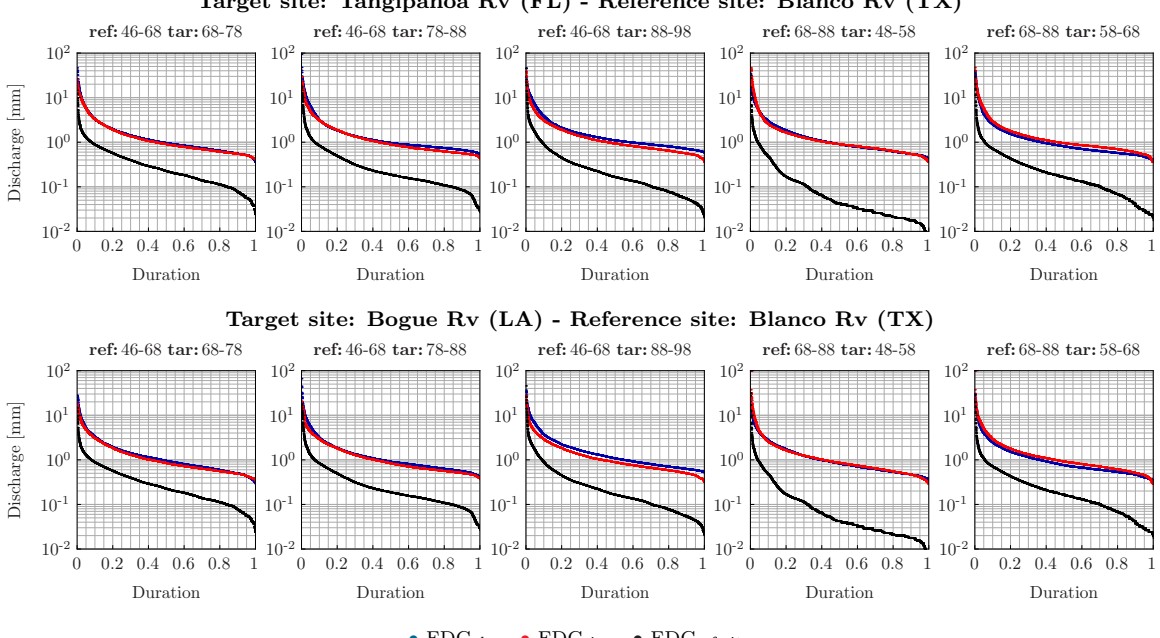

**Figure 5.** Simulated FDC at the target catchments Tangipahoa River (FL) and Bogue River (LA), upper and lower panel, respectively, from the data of the reference catchment Blanco River (TX). The reference years are a 20 years time window from October 1948 to September 1968 (left panels) and from October 1968 to September 1988 (right panel). Target years are the decades from October 1968 to September 1998. The blue dots are the observed FDC at the target catchment; the red dots are simulated FDC at the target catchment and the black dots are the observed FDC at the reference catchment.

It is possible to conclude that the API gives effectively a good estimation of the memory of the basin and can be used to represent the precipitation similarly to the discharge.

Moreover, these results show that the FDC is not a characteristic of the catchment properties only, but of both the catchment and the weather. To better investigate these findings, we performed the KS test on pairs of observed and simulated FDCs for two purposes. The first is to know if pairs of simulated and observed FCDs at the same site have the same continuous distribution,
5 the second is to know which is the distance between the pairs. As mentioned in Section 3.1, the test rejected the null hypothesis that FDCs built in different time periods had the same distribution. The test performed on pairs of simulated and observed FCDs revealed that in 48% of the cases, for Tangipahoa River, the null hypothesis could not be rejected. Moreover, we wanted to know which is the distance between pairs of FDCs built at the same site in different periods of time. Then, this distance was compared with the distance between simulated and observed FCDs at the same period of time. Results shown that the
10 distance between the former pairs is bigger than the distance between the latter, Figure 9. This means that the methodology proposed here has a good performance and it provides an interesting alternative to other methodologies which assume that FDC of different periods of time have the same distribution.



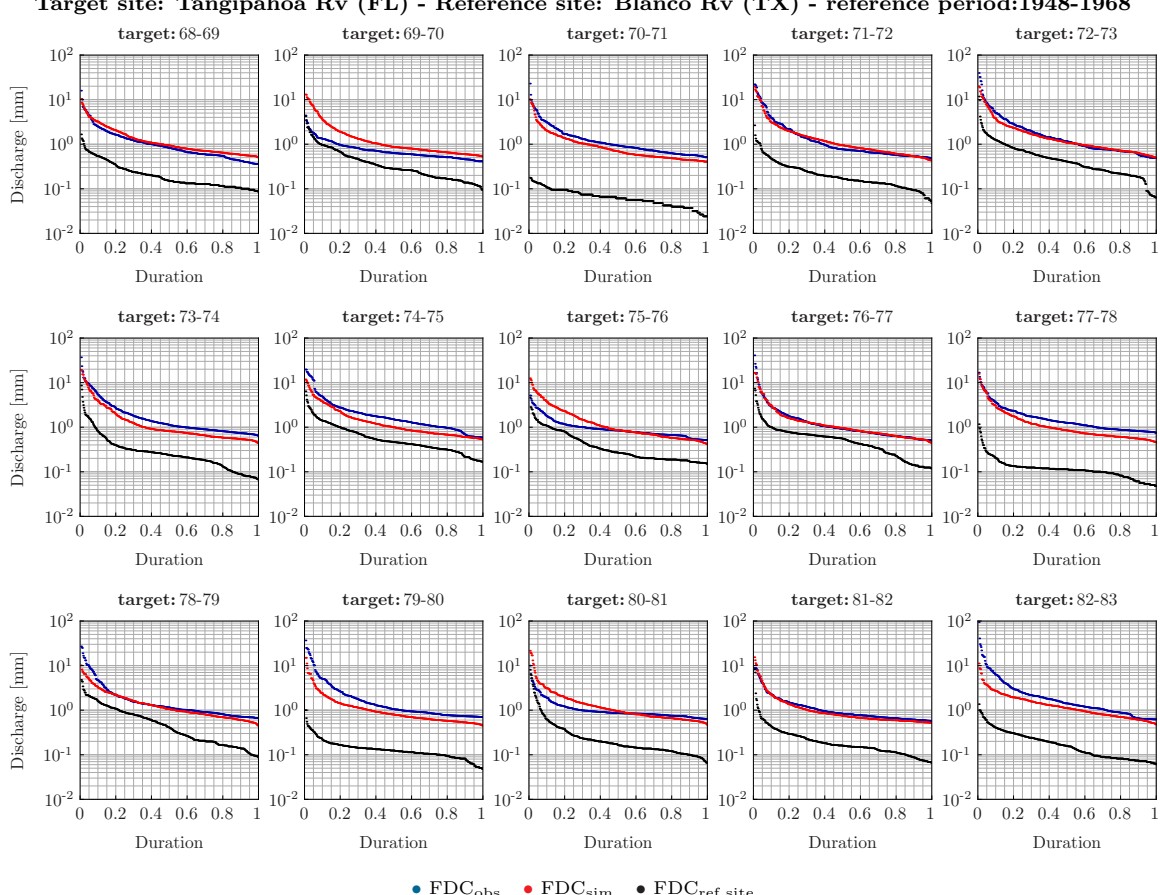

**Figure 6.** Simulated FDC at the target catchment Tangipahoa River (FL) from the data of the reference catchment Blanco River (TX). The reference years are a 20 years time window from October 1948 to September 1968. Target years are each year from October 1968 to September 1982. The blue dots are the observed FDC at the target catchment; the red dots are simulated FDC at the target catchment and the black dots are the observed FDC at the reference catchment.

Since we found that the weather conditions strongly influence the FDCs estimation, we analysed also the streamflow percentiles to assess the between-year variability. To this end, the moving average (MA) of 30th, 70th, 90th and 99th percentiles of streamflow is estimated. The MA values are estimated using three different fixed time windows (i.e., 10, 15 and 20 years), Figure 10. It is interesting to observe that the MA values are characterized by a strong variability throughout the time. The fluctuation of the flow percentiles suggests that the percentiles cannot be considered an invariant characteristic of the basin.

5 Therefore, it is not possible to estimate the flow quantiles using regression methods that do not take into account the weather characteristics. These methods, first, regionalise empirical runoff percentiles using multiple regression models. Then, regional evaluation of flow percentiles are interpolated across the percentiles (e.g., Franchini and Suppo, 1996; Smakhtin, 2001). If flow percentiles are estimated separately from weather characteristics, it may results in a misrepresentation of the percentiles





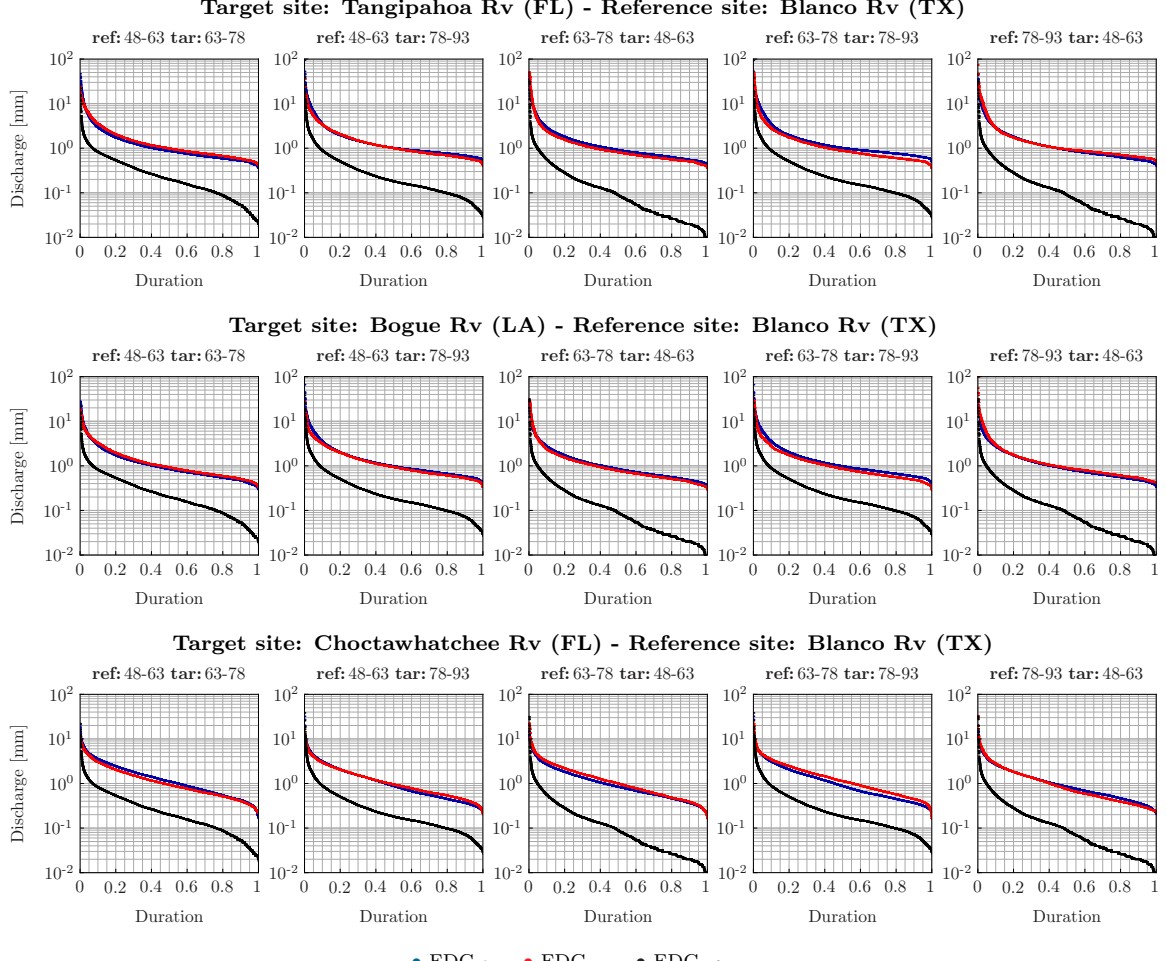

**Figure 7.** Simulated FDC at the target catchments Tangipahoa River (FL), Bogue River (LA) and Choctawhatchee River (FL), upper, middle and lower panels, respectively, from the data of the reference catchment Blanco River (TX). The reference and target years are periods of 15 years. The blue dots are the observed FDC at the target catchment; the red dots are simulated FDC at the target catchment and the black dots are the observed FDC at the reference catchment.

themselves. Therefore, we suggest to add a weather factor to take into account the influence of the weather in the percentiles estimates.

## 4.1 Performance criteria

To determine the performance of the procedure proposed in this paper, different criteria are selected: the Nash-Sutcliff efficiency index, the BIAS and the mean absolute error (MAE).




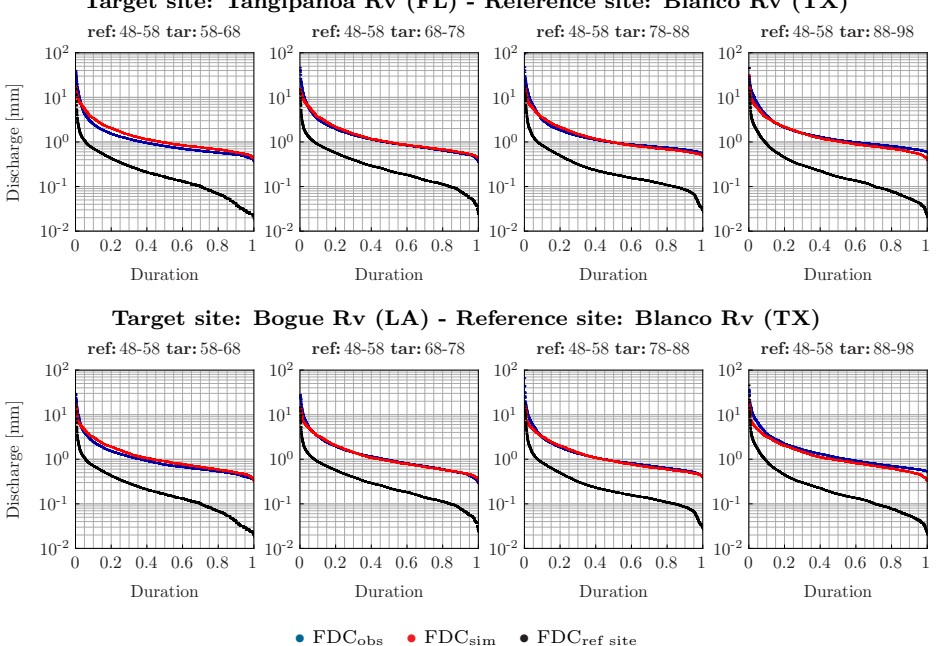

**Figure 8.** Simulated FDC at the target catchments Tangipahoa River (FL) and Bogue River (LA), upper and lower panel, respectively, from the data of the reference catchment Blanco River (TX). The reference and target years are periods of 10 years. The blue dots are the observed FDC at the target catchment; the red dots are simulated FDC at the target catchment and the black dots are the observed FDC at the reference catchment.

The Nash-Sutcliffe efficiency between the simulated and the observed flow value is the most widespread performance criterion:

$$\text{NSE} = 1 - \frac{\sum\limits_{i=1}^{N}\left(Q_{\text{obs},i} - Q_{\text{sim},i}\right)^2}{\sum\limits_{i=1}^{N}\left(Q_{\text{obs},i} - \bar{Q}\right)^2}, \tag{4}$$

where $Q_{\text{obs}}$ is the observed discharge value at the target catchment during the target period; $\bar{Q}$ is the mean value of the observed discharge during the target period in the target catchment; $Q_{\text{sim}}$ is the simulated discharge value. The NSE is evaluated here for a specific set of percentiles, thus, $N$ is the number of discharge values related to a specific percentile.

The BIAS represents the mean difference between observed and simulated values:

$$\text{BIAS} = \frac{1}{N} \sum\limits_{i=1}^{N} \left(\frac{Q_{\text{obs},i} - Q_{\text{sim},i}}{Q_{\text{sim},i}}\right). \tag{5}$$





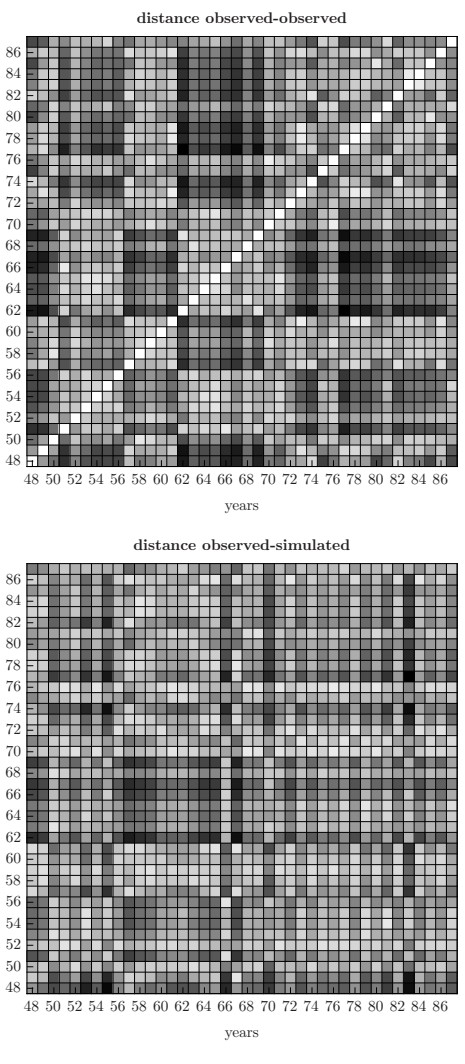

**Figure 9.** Kolmogorov-Smirnov distance between couples of streamflow values observed (upper panel) and between couples of streamflow values observed and estimated (lower panel) at Tangipahoa River (FL) from October 1948 to September 1987.

If the BIAS equals zero there is a perfect fit between observed and simulated values. If the BIAS is negative, observed values are overestimated, while if the BIAS is positive, they are underestimated.

The mean absolute error is defined as:

$$\text{MAE} = \frac{\sum\limits_{i=1}^{N} |Q_{\text{obs},i} - Q_{\text{sim},i}|}{N}. \tag{6}$$





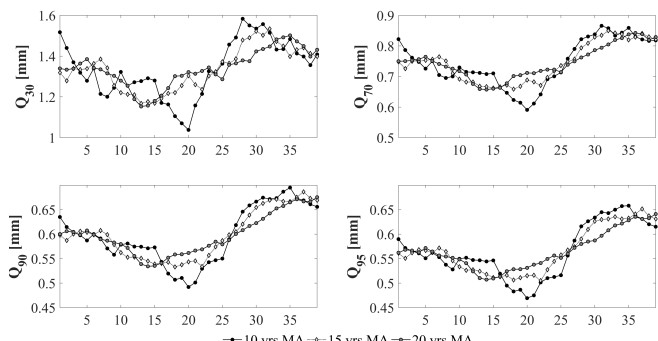

**Figure 10.** Moving average (MA) of the 30th, 70th, 90th and 99th percentiles of daily streamflow values gauged at Tangipahoa. Three different fixed time windows are used to estimate the MA: 10, 15 and 20 years.

It measures the overall agreement between observed and simulated values. It is a non-negative metric without upper or lower bounds. A perfect model would result in a MAE equals to zero. This estimation metric does not provide any information about under- or over-estimation, but it determines all deviations from the observed values regardless of the sign.

For the German catchments the Ratio, the Mean Error (ME) and the Volume Error (VE) are estimated. The Ratio is defined as:

$$5 \quad \text{Ratio} = \sqrt{\frac{\sum\limits_{i=1}^{N} \left(Q_{\text{obs},i} - Q_{\text{sim},i}\right)^2}{\sum\limits_{i=1}^{N} \left(Q_{\text{obs},i} - \bar{Q}\right)}}. \tag{7}$$

The value of the ratio equals zero when the model has a high performance. The Mean Error (ME) is given by:

$$\text{ME} = \frac{\sum\limits_{i=1}^{N} Q_{\text{sim},i} - Q_{\text{obs},i}}{N}. \tag{8}$$

The optimum value of the ME is zero. If the ME value is close to zero, the model simulation is accurate. If the value of the bias is lower than zero, ME shows an underestimation of the observed data. otherwise, the ME shows an overestimation of the data.

The Volume Error is defined as the ratio between the difference between the total simulated data and total observed data and the total observed data:

$$\text{VE} = \frac{\sum\limits_{i=1}^{N} Q_{\text{sim},i} - \sum\limits_{i=1}^{N} Q_{\text{obs},i}}{\sum\limits_{i=1}^{N} Q_{\text{obs},i}}. \tag{9}$$



If the VE value is close to zero, the accuracy of the model is high.

For a deep understanding of the methodology goodness, the estimation metrics are evaluated for the 1st, 3rd, 5th, 10th, 20th, 30th, 50th, 75th, 90th and 99th percentiles.

For U.S. catchments, when a decade is used as both target and reference period, the performance measures show a good agreement between observed and simulated values, Figures 11.

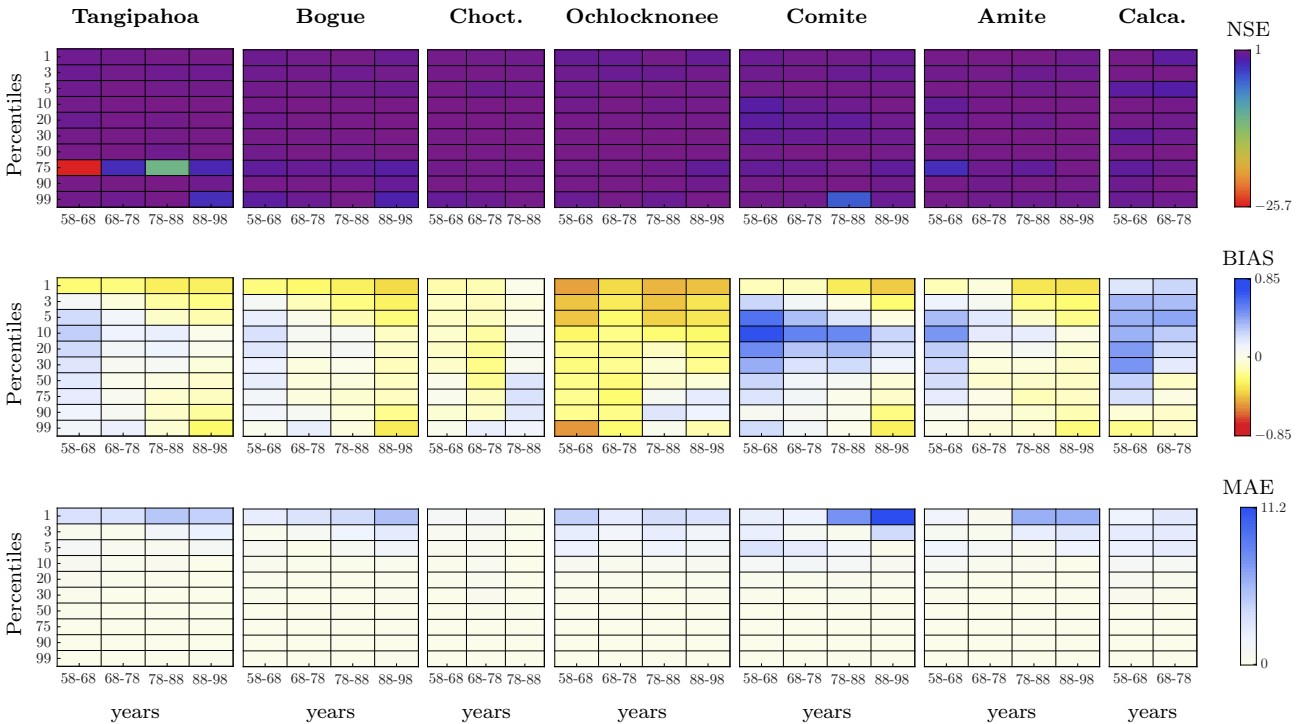

**Figure 11.** Performance measures NSE, BIAS and MAE evaluated for specific percentiles (on the y-axis) and for specific target decades (i.e., 1958-1968, 1968-1978, 1978-1988 and 1988-1998 on the x-axis). The reference decade is 1948-1958, the reference catchment is Blanco (TX). Each target catchment is indicated above the corresponding box.

5     The NSE index is characterized by values close to 1 especially for intermediate percentiles. Also the BIAS and the MAE shows a low performance for high streamflow values (i.e., low percentiles). This is due to the fact that the procedure is more able to reproduce the average streamflow values than extreme events such as floodings and droughts. This is in agreement with what was already observed in Figure 8. The same behaviour can be observed when both target and reference periods equal 15 years, Figure 12.

10   For the German case study, errors are reported for Wannweil, Horb and Oberensingen Aich as target catchments, using as reference catchment Plochingen, Neckar. Three different cases are shown: a target time period of 20 years (i.e., from 1961 to 1980) and a reference time period of 10 years (i.e., from 1981 to 1990) are used as Test 1. In Test 2, both the target and the reference time periods equal 15 years, i.e., from 1961 to 1975 and from 1976 to 1990 as target and reference, respectively. For





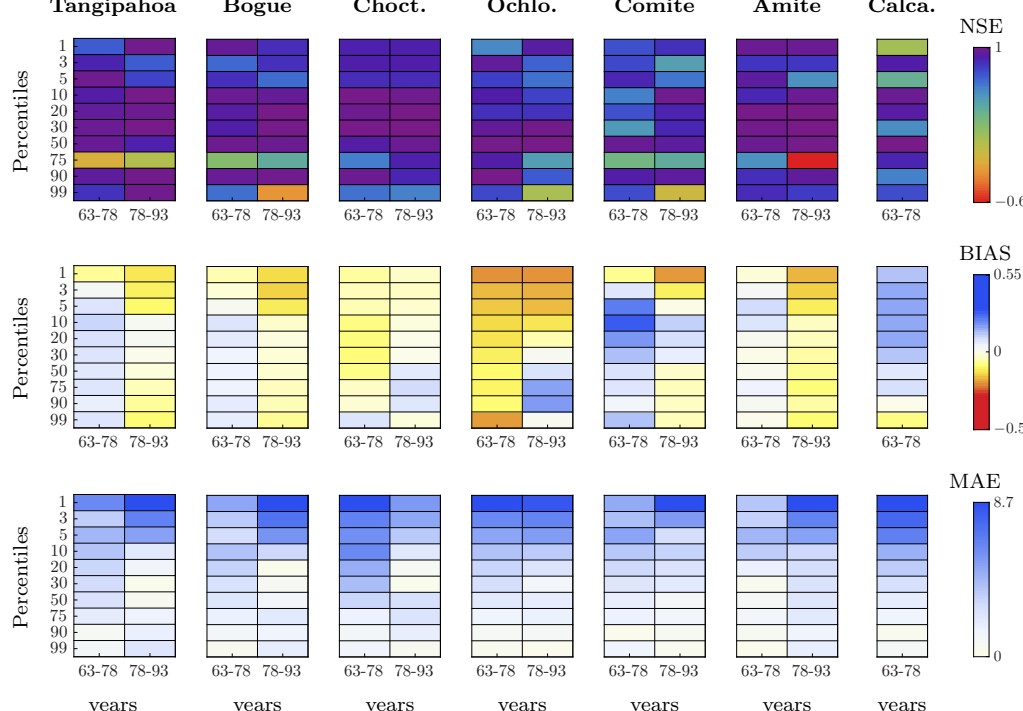

**Figure 12.** Performance measures NSE, BIAS and MAE evaluated for specific percentiles (on the y-axis) and for a specific 15 years target (i.e., 1963-1978 and 1978-1993 on the x-axis). The reference decade is 1948-1963, the reference catchment is Blanco (TX). Each target catchment is indicated above the corresponding box.

Test 3 the target time period is of 10 years (i.e., from 1961 to 1970) and the reference time period is of 20 years (i.e., from 1971 to 1990), Figure 13.

As for the U.S. catchments, for Wannweil and Horb, the NSE shows values close to one for intermediate percentiles, while the performance decreases for the extreme percentiles. While for Oberensingen Aich, the intermediate percentiles shows a poor performance. These performances are confirmed also by the Ratio index. However, the error values given by the ME and the

5    VE are close to zero in the majority of the cases. The errors increase for the percentiles corresponding to floodings for Horb and Oberensingen Aich.

To better understand the relationship between a target and a reference catchment, the coefficient of correlation has been computed. Coefficient values are reported for Test 1. Values are estimated between the reference catchment Plochingen Neckar and four target catchments: Wannweil, Riederich, Plochingen Fils and Horb, Figure 14. The coefficient shows values close to

10   1 for almost all the percentiles and for all catchments. The smallest coefficient values correspond to high streamflow values for Horb and to droughts for Plochingen Fils. Plochingen Fils behaviour is not well represented. This can also due to the fact that Plochigen Fils is characterized by a karstic area as shown also by Samaniego et al. (2010). The low correlation existing





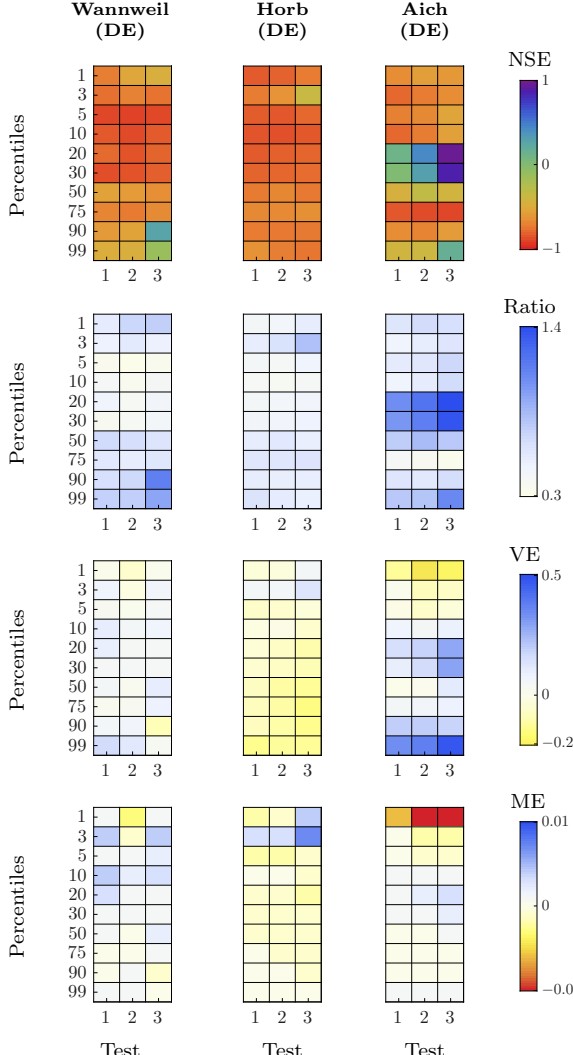

**Figure 13.** Performance measures NSE, Ratio, Volume Error (VE) and Mean Error (ME) evaluated for specific percentiles (on the y-axis) and for three specific set of target and reference years (i.e., Test 1, 2 and 3). For Test 1 the target time period is from 1961 to 1980 and the reference time period is from 1981 to 1990. For Test 2 the target time period is from 1961 to 1975 and the reference time period is from 1976 to 1990. For Test 3 the target time period is from 1961 to 1970 and the reference time period is from 1971 to 1990. The reference catchment is Plochingen Neckar, while the target catchments are Wannweil, Horb and Oberingen Aich, as indicated on each corresponding box.

between Plochingen Neckar and Horb results in a poor performance when the data of Plochingen Neckar are used to simulate Horb. Indeed, the error at the corresponding percentile is high (e.g., ME, Figure 13).



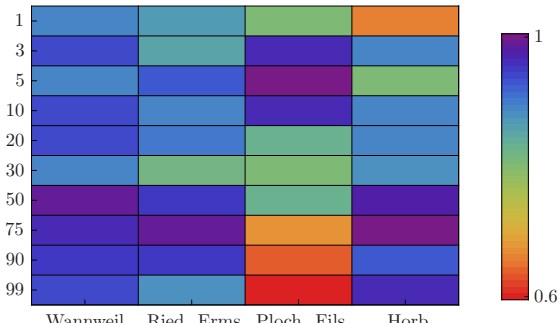

**Figure 14.** Correlation coefficient evaluated at each percentile between Plochingen Neckar and each other catchments indicated on the x-axis.

## 5 Conclusions

The paper presents a new methodology to estimate the streamflow behaviour at ungauged and partially gauged catchments, given the discharge and the precipitation gauged at another catchment. Here it is shown that two FDCs built for the same catchment, but with data corresponding to two different time windows, cannot be regarded as the same continuous distribution. The same results when two FDCs of two different catchments built for the same time window are analysed. Thus, it is

not possible to infer a FDC using parameters retrieved from the distribution of another FDC without considering the weather. The FDC at a specific site is not a property of the corresponding basin, but the FDC is a property of both the basin and the weather. The influence of the weather is evident analysing the between-year variability of flow percentiles. Indeed, the moving average of the 30th, 70th, 90th and 99th flow percentiles shows a strong variability throughout the time. This behaviour has a strong consequence as it means that it is not possible to retrieve the streamflow percentiles without taking into account

the weather. Indeed, there exists several methodologies (i.e., regression models) that estimate flow quantiles separately from weather characteristics. FDCs and their selected properties cannot be considered as catchment characteristics and should be used with caution for regionalization purposes.

Because of the dependence on the climate, discharge data are here retrieved using the precipitation data series. Since precipitation data series are characterized by a high number of zeros, here we used the Antecedent Precipitation Index (API). The API

is determined from the precipitation and allows us to exploit the precipitation in a similar way to discharge data. Indeed, the API is an index for soil moisture condition, it is related to the amount of water released to the river and thus, to the discharge.

The streamflow data and thus, the FDC at a target site are determined for a specific time window (i.e., target period) using API and discharge available for a so-called reference time period at another catchment (i.e., reference site).

Moreover, we tested a method to retrieve the FDC of a partially ungauged site from the FDC of a gauged one.

FDCs simulated at the target site are compared with FDCs that were actually observed at the same site. To test the methodology, several reference periods (i.e., for which streamflow data are available) and target ones (i.e., for which streamflow data are to be determined) are analysed, such as 1 year, 10, 15 and 20 years and two case study areas are investigated. From the comparison of observed and simulated FDCs, it results that the methodology is able to correctly determine the missing streamflow data. To exploit the methodology goodness several estimation metrics are computed and analysed. It is note worthy that the





discharge values of the intermediate percentiles are better described than those of the extremes. Nevertheless, the error values between observed and simulated FDCs are small. Moreover, it is interesting to observe that simulated and observed FDCs are similar in shape, even when the two curves are not overlapping. This behaviour can be due to the different temperature values characterizing the two catchments. Indeed, a high difference in temperature can cause a different evapotranspiration which in turn can influence the discharge. To better analyse the relationship between reference and target catchments, the co-

efficient of correlation is computed between discharge data gauged at the two sites of interest during the reference period. As the performance criteria highlighted, the data series are more related at the intermediate percentiles and less at the extremes. The correlation coefficient estimated for the reference period can help to determine in advance whether the discharge data of a reference and a target catchments are strongly correlated during that period of time. The FDCs estimated at the target site will be more accurate if the correlation coefficient shows a strong correlation. This could be useful especially if no data are actually

available at the target site for the target period and it is not possible to estimate the goodness of the simulated FDC.

The catchments used for analysis here are characterized by a mild climate, thus the snow melt effect was not taken into account. However, if the methodology would be applied to catchments characterized by snowfalls, the snow melt effect should be taken into account adding the snow melt to the API.

**Appendix A**

In this Appendix we want to provide an easy example to better understand the method that we applied for U.S. catchments. This method is based on the use of the API. We recall that a "reference period" is a period of time for which streamflow values are available at both reference and target catchments, while a "target period" is a period of time during which streamflow values are not available at the target catchment. API values are available at both catchments for both periods of time.

Let suppose that we want to know the discharge value at catchment B (i.e., Bogue Rv, LA) corresponding to the 10.11th

percentile (i.e., 10.11%) for the year ranging from October 1968 to September 1969. Let suppose that the reference period has a length of 15 years and ranges from October 1948 to September 1963. We present the method step by step in the following.

    1. Select the mean daily precipitation occurred at the reference catchment (i.e., Blanco Rv) during the target period and estimate the API as in Eq.3 assuming $\alpha$ equal to 0.85;

    2. sort in descending order the API values evaluated for the target period at the reference catchment (i.e., Blanco Rv, TX);

3. assign to each sorted value the corresponding rank $i$, with $i = 1,..., N_t$ where $N_t$ is the length of the target API series and thus equals 365, and then estimate the exceedance probability $P(Q < q_i)$ of each value using a Weibull plotting position $i/(N_t + 1)$, Table A1;

    4. in the sorted API series, identify the value with frequency equal to 10.11%. This value equals 37.72 mm (bold line in Table A1);



5. estimate the API from the mean daily precipitation occurred during the reference period at the reference catchment (i.e., Blanco Rv, TX) and sort in descending order the API values, estimate the rank and the associated exceedance probability $P(Q < q_i)$ of each value as $i/(N_r + 1)$ where $N_r$ equals 5475;

6. find the exceedance probability $P(Q < q_i)$ associated to the value 37.72 mm in the sorted API sample. From Table A2 it is possible to observe that there is not such an API value. Therefore, look for the two most similar values: one should be bigger and the other smaller than the searched value. Then, take their empirical frequency values (i.e., 7.52 % and 7.54%; in bold, Table A2);

7. sort in descending order the streamflow values gauged during the reference period at the target catchment (i.e., Bogue Rv, LA), estimate the rank and the associated exceedance probability $P(Q < q_i)$ of each value as $i/(N_r + 1)$;

8. find the two streamflow values which have an empirical frequency equal to 7.52 % and 7.54%. These values are in bold, Table A3;

9. estimate the mean value of these two streamflow values. The resulting value is the streamflow value with empirical frequency equal to 10.11% evaluated for the target catchment and the target period that we were looking for, Table A4.



**Table A1.** API values sorted in descending order and the corresponding percentiles estimated for the target year (i.e., 1968-1969) at the reference catchment (i.e., Blanco RV, TX).

| Rank | $P(Q < q_i)$ | $API_{Blanco,tar}$ |
|------|------|------|
|      | % | mm |
| 1 | 0.27 | 76.78 |
| 2 | 0.55 | 73.39 |
| … | … | … |
| 30 | 8.20 | 39.65 |
| 31 | 8.47 | 39.35 |
| 32 | 8.74 | 38.71 |
| 33 | 9.02 | 38.31 |
| 34 | 9.29 | 38.18 |
| 35 | 9.56 | 38.10 |
| 36 | 9.84 | 37.97 |
| 37 | **10.11** | **37.72** |
| 38 | 10.38 | 36.99 |
| … | … | … |
| 365 | 99.73 | 0.61 |





**Table A2.** API values corresponding to specific percentiles estimated for the reference years (i.e., 1948-1963) at the reference catchment (i.e., Blanco RV, TX).

| Rank | $P(Q < q_i)$ % | $API_{Blanco,ref}$ mm |
|---|---|---|
| 1 | 0.02 | 266.17 |
| … | … | … |
| 410 | 7.49 | 37.81 |
| 411 | 7.51 | 37.78 |
| 412 | **7.52** | **37.74** |
| 413 | **7.54** | **37.61** |
| 414 | 7.56 | 37.61 |
| 415 | 7.58 | 37.55 |
| … | … | … |
| 5475 | 99.98 | 0.01 |





**Table A3.** Streamflow values corresponding to specific percentiles gauged during the reference years (i.e., 1948-1963) at the target catchment (i.e., Bogue RV, LA).

| Rank | $P(Q < q_i)$ %  | $q_{Bogue,ref}$ mm |
|------|------|------|
| 1 | 0.02 | 38.81 |
| … | … | … |
| 410 | 7.49 | 3.28 |
| 411 | 7.51 | 3.28 |
| 412 | **7.52** | **3.21** |
| 413 | **7.54** | **3.21** |
| 414 | 7.56 | 3.20 |
| 415 | 7.57 | 3.20 |
| … | … | … |
| 5475 | 99.98 | 0.31 |





**Table A4.** Streamflow value corresponding to the 10.11th percentile estimated for the target year (i.e., 1968-1969) at the target catchment (i.e., Bogue RV, LA).

| $P(Q < q_i)$ | $q_{Bogue,tar}$ |
| --- | --- |
| % | mm |
| 10.11 | 3.21 |

*Competing interests.* No competing interests are present.





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
