# Peer review of "A methodology to estimate flow duration curves at partially ungauged basins 1"

_Hydrology and Earth System Sciences, 2018_

## Referee Comment (RC1) · W.H. Farmer (Referee) · 20 Jul 2018

The authors have provided an interesting hypothesis that a daily flow duration curve, as function of both basin characteristics and climate, can be reproduced with the use of a precipitation index. While this is an intriguing assertion, the experiment and manuscript need some revision before being published. A previous commenter has noted some concerns, and I will try to expand on those concerns here with an aim towards supporting revisions of this work.

This work could benefit from a clear statement of hypotheses. In my opinion, the main hypothesis is that the daily flow duration curves at an ungauged location can be simulated with knowledge of the precipitation record at both the ungauged site and

some index site. This hypothesis relies on a further assumption that the cumulative distributions of streamflow and precipitation correlate in some way. In the revisions I am proposing, I think the authors should clearly set out to quantifiably address these hypotheses.

The methods section needs a great amount of revision. A figure may improve the understanding of the methodology. As a previous commenter noted, the approach is difficult to understand and may be greatly simplified. The authors create cumulative distribution functions (CDFs) for streamflow and API at both an index site and a target site over some reference period. They then create a CDF of streamflow and API at the index site for some target period and a CDF for API at the target site for this same target period. The method then only uses the CDFs of (1) API at the index site in the target period, (2) API at the index site in the reference period, and (3) streamflow at the target site in the reference period.

In addition to improving readability, revising the methods section might also address the concerns raised by the previous commenter. Namely, that it seems the approach could be greatly simplified through interpolation along the relevant relationships without the need for intermediate exceedance probabilities. This would be accomplished by the following: (1) Create the CDF of API at the index site in the target period, (2) Plot the API of the index site in the reference period against the streamflow at the target site during the reference period, (3) interpolate each API from the target period, in order, along the curve created in (2) to produce the CDF of streamflow at the target site in the target period. While this approach is achievable algorithmically, and identical to the one proposed, it raises several concerns about the implicit assumptions.

Are API and streamflow ranked independently? (See step 3 of page 10.) If so, the implicit assumption is that the exceedance probability of the API on a given day is equivalent to the exceedance probability of the streamflow on that same day at the same site. This is a pretty sizeable assumption. As a start, it would be good to see if the temporal sequence of API exceedance probabilities is highly correlated with the

temporal sequence of streamflow exceedance probabilities at a single site over the same period.

The second assumption arises when we move from the CDF of API at the index site in the reference period to the CDF of streamflow at the target site in the reference period. This movement introduces a second implicit assumption: namely, that the exceedance probability of the API on a given day at the index site is equivalent to the exceedance probability of streamflow on that same day at the target site. Put another way, if you accept the assumption in the previous paragraph, this step assumes that the temporal sequencing of API is identical at both sites. Again, this needs to be demonstrated: Is the temporal sequence of API exceedance probabilities at the index site highly correlated with the temporal sequence of streamflow exceedance probabilities at the target site in the reference period?

It may be argued that the temporal sequencing is irrelevant. This is not the case. By assuming the same exceedance probabilities in step 7 of page 10, we are assuming a perfect correlation and, therefore, assuming a temporal correspondence.

The third assumption, which was alluded to earlier, is that the CDF of API is identical across sites for both the index site and the target site in the same period. This is what allows the authors to use the CDF of the API of the index site in the target period for step 6 on page 10. It may be that this is what the authors meant by "the assumption of large scale precipitation" (line 16, page 9); if so, please clarify. Regardless, this assumption needs to be validated through correlation or a KS test.

Without some quantifiable validation of these assumptions, the proposed method is tenuous at best and left vulnerable to criticism. With that in mind, and the comments of the previous commenter, I'd like to propose that exploring these assumptions might result in modifications of the methodology that might move away from the case of simple interpolation. Is the relationship between API and streamflow constant across periods or sites? Should API and streamflow be ranked independently or with some sort of

dependence? Should the API of the index site in the target period be used to map to a different site in a different period (i.e., the target site in the reference period)? Exploring these questions, and validating the underlying assumptions, will produce a more robust approach.

In addition to their main hypotheses proposing this methodology, the authors assert that the FDC is a product of the basin and the weather. This is surely intuitive, but the evidence provided could be greatly strengthened. The authors use KS tests, but is unclear how they were applied. It would be informative to clearly communicate if the CDF of streamflow from one period and the CDF of streamflow from another period could be considered significantly different. The authors have done this, but the presentation is not clear. The extension would be to ask if the API can be correlated with any differences across time. (As an aside: Was there any discussion of selecting stationary sites? How would nonstationary behavior play a role here?)

This, in my opinion, raises another concern: The authors seem to be attempting to simultaneously address two very different problems. The first problem considers a target site that has a streamflow record overlapping with an index site, but the desired period has no overlap (the ungauged area is the same site, different period). In this case, the use of APIs within site, without an index site, would be most ideal. The second problem considers a site without any streamflow information; this situation necessitates the use of an index. Of course, when there are gaps in the API record as well, this transforms into four unique problems. Regardless, if we believe the underlying assumption that the CDF of streamflow is a product of basin and weather, then the solutions to these problems must be quite different. The first asks if knowledge of new weather can produce the CDF of streamflow, while the second alters both variables and asks if the CDF relationship can be transferred across weather and basin. Line 8 of page 3 implies that both problems are considered, but the remainder of the paper seems only to address the partially gauged site. I would advise addition of the second problem or, at least, a discussion of implication for the second problem (completely ungauged).

Before moving to some more specific comments, I would like to direct the authors to a couple previous works that should probably be discussed. In 1996, Hughes and Smakhtin (<https://doi.org/10.1080/02626669609491555>), among others, provided a technique for hydrograph simulation using flow duration curves. While their focus was on hydrographs, the extensions to ungauged FDCs can be made quite clearly (i.e., they could be derived from simulated hydrographs). Smakhtin and Masse (2000: <https://doi.org/10.1002/(SICI)1099-1085(20000430)14:6%3C1083::AID-HYP998%3E3.0.CO;2-2>) then extended this method to use a precipitation index. While I believe that the methods presented here are different, the novelty of this new method must be strongly articulated.

SOME MORE SPECIFIC COMMENTS:

I strongly encourage the authors to revisit the style of the manuscript. At times, it feels a bit disjointed and it may be improved by enforcing a strict Introduction-Methods-Results-and-Discussion (IMRAD) format. For example, section 3.1 is ostensibly a methods sections but presents a series of results that I think are pivotal to the paper (line 19, page 8). Similarly, the paragraph on page 13 and section 4.1 present new methods of analysis that have not been presented earlier in the paper. While IMRAD is not a requirement, I do suggest thinking carefully about the best approach to presenting the narrative.

In my opinion, this work needs more presentation and discussion of quantified results. The results sections heavily rely on visualization. Even the presentation of metrics in section 4.1 is visual. While this is useful, we still need to see some discussion on the performance metrics. For example, the scale on NSE in Figure 11 makes all positive values appear as a single color. This presentation means we can't honestly see how the methods perform.

Page 1, line 17: When talking about general duration curves, more commonly known as cumulative distribution functions, it is better to say "exceedance frequency" rather

than "exceedance time".

Page 2, line 1: Please provide the citation for the Weibull plotting position.

Page 3, line 4: Please provide more discussion and literature of this important point.

Page 3, line 7: It is not clear what the "distribution of the FDC" is. The FDC is a distribution, so it is confusing to talking about the distribution of a distribution.

Page 4, line 6: Florida, Louisiana and Texas are certainly not the East coast. I would suggest the Gulf Coast.

Page 5, line 7: misspelling of database

Page 5, line 2 (?): This is an example of inconsistent citation style. Bloeschl should be in parenthesis.

Page 8, line 5: Please provide citation to KS test.

Page 9, line 10: What lead to this choice for alpha? (Also, note that the same symbol is used earlier in this section for significance: page 8, line 10.) Please provide citation or summary of initial exploration.

Page 9, line 18: I strongly suggest referring to the "reference site" as an "index" or "donor". The reference connotation implies lack of human influence that might be confusing. The same could be said of the reference period.

Page 10, line 17: The series of $N_r$ and $N_t$ are both being indexed with i, which leads to confusion.

Page 10, line 19: So, $API\_A_{ti}$ is equal to $API\_A_{rj}$?

Page 11, line 4: What is a supporting variable? This is not described as such earlier, so it surprises the reader.

Page 11, line 17: "good agreement" is very subjective. Please provide thorough, quantifiable analysis. For example, a lot of the curves in figure 6 look rather poor for highs

and lows (top row, second box from the left).

Figure 5: Why was the box for ref:68-88 and tar: 88-98 not included? The caption needs to do a better job of describing the different panels.

Page 12, line 9: Spelling of FDCs.

Page 13: The methods for this paragraph were very unclear to me. Could a figure or a revision help?

Page 14: Please provide the citation for NSE. Even better, a metric like KGE might be more appropriate.

Figure 10: What is the horizontal axis of this figure?

CLOSING REMARKS:

I want to express my deepest appreciation for this work and convey my strongest encouragement. I think the topic is quite relevant and the hypotheses are very intriguing. With some additional redevelopment, I think this work could address some fundamental questions of hydrology. It is for this reason that I invested so much time in the review. While some of my conclusions might be inaccurate, I hope I have provided some food for thought as to improving this work. If you have any concerns, please do not hesitate to reach out to me again. This is a topic I'd be more than happy to discuss at length. Thank you.

William Farmer

---

## Short Comment (SC1) · 20 Jul 2018

Dear Elena,

This is a very quick comment about your algorithm. Please note that, in principle, you do not need to introduce any probability Pi and Pj (leaving aside that the standardized ranks of nonhomogeneous seasonal flow are not estimators of probabilities). In fact, your sorted API_Ar and q_Br define a monotonic function f : API_Ar -> q_Br such that new/unknown values q_Bt can be found by simple interpolation. Their standardized ranks are not needed for interpolation, and those of the sorted q_Bt are automatically defined by the position of each value in the sorted target sequence.

What you do can easily be performed without nuisance (I would say, superfluous) prob-

abilites by the R function 'approx' (see codes below) or whatever similar interpolation function in other software. By the way, instead of using API values corresponding to 'probability' values Pj and Pj+1 when the target value is not available in the reference set, a better approach is the linear interpolation (which is what 'approx' and whatever standard interpolator does). This is standard when dealing fractional order statistics, which are actually the objects you are handling in this work.

Moreover, interpolation works if the range API_Ar includes all the values of API_At. Otherwise, extrapolation is needed, and it can be problematic if we rely on empirical survival functions. Another minor comment: you state 'Specifically, the procedure involves daily Antecedent Precipitation Index (API) and daily streamflow values of both A and B.' I may have missed something, but from your example, it seems that you need only API in A for the reference and target period, and flow q in B for the reference period.

However, my main concern is about the justification of the procedure, i.e. the link between weather variables (e.g. API) and flow. Since you work with sorted sequences, when you arrange API and q in ascending order, you lose the correlation between simultaneous pairs (API, q). This means that the physical link between API and q (...showing the correlation/scatter plots of such variables could be useful) does not play any role in your procedure. The simple R code below show how we can closely reproduce your results dealing with perfectly independent or dependent random variables X and Y. I used three setups: (1) a trivial iid, (2) a scenario where I introduced level shifts for groups of 365 realizations, mimicking the possible (and generally evident) fluctuations of the annual average flow (in the spirit of the index-flow flow duration curves of Castellarin et al.), and (3) a third case with shifts and some correlation. Running the code a few times, in all cases, I obtained results close to yours, even though these realizations are purely fictional. Of course, you can play with this code changing parameters, sample size, distributions, etc.

However, my message is to explore a bit more what you are doing and why you obtain

those results. To be honest, I think that whatever variable recorded in the reference and target period can be used as a support for interpolation, if its realizations in the reference period explore the sample space in the same way as the realizations of the target variable do. What we see in your diagrams concerns the properties of order statistics and sampling rather than physics. In any case, you should perform extensive simulations under controlled conditions to support your assumptions, in order to understand which parameter/aspect matters. Please, remember that our algorithms should be falsified rather than verified. Another way to check your results is to use other random variables as a reference, e.g. pure fictional variables, weather/climate indices showing weak or null correlation with flow, etc.

Minor remark: you use 'target site', 'reference site', 'target period', and 'reference period'. This can be confusing. I can suggest 'donor site' and 'ungauged site'. Finally, 'simulated' values in the target period can be denoted more properly as 'interpolated' values I think, because 'simulated' usually refers to some pseudo-random generation or stuff like that, which is not what is done here.

As ever

F

PS: if you need further information, discussion, etc., feel free to mail me.

```
#############################################
set.seed(666)

xa.r = rnorm(365*20)

yb.r = rlnorm(365*20)

xa.t = rnorm(365)

yb.t.obs = rlnorm(365)
```

```
yb.t.sim = approx(x = sort(xa.r), y = sort(yb.r), xout = sort(xa.t))$y

plot(1-ppoints(365,0),sort(yb.t.obs) , type="l", log="y") lines(1-ppoints(365,0), yb.t.sim
,col=2)

**with perturbation mimicking fluctuations of annual mean levels**

set.seed(6666)

xa.r = rnorm(365*20)+rep(rnorm(20,sd=0.2),each=365)

yb.r = rlnorm(365*20)*rep(rlnorm(20,sd=0.6),each=365)

xa.t = rnorm(365)

yb.t.obs = rlnorm(365)

yb.t.sim = approx(x = sort(xa.r), y = sort(yb.r), xout = sort(xa.t))$y

plot(1-ppoints(365,0),sort(yb.t.obs) , type="l", log="y")

lines(1-ppoints(365,0), yb.t.sim ,col=2)

**with perturbation and some cross-correlation**

xa.r = rnorm(365*20)+rep(rnorm(20,sd=0.4),each=365)

yb.r = rlnorm(365*20)*rep(rlnorm(20,sd=0.6),each=365)

rho = 0.8

corM <- rbind(c(1.0, rho), c(rho, 1.0))

SigmaEV <- eigen(corM)

Meps <- rbind(xa.r,log(yb.r))

Meps <- SigmaEV$vectors %*% diag(sqrt(SigmaEV$values)) %*% Meps

Meps <- t(Meps)
```

```
**plot(Meps)**

xa.r = Meps[,1]

yb.r = exp(Meps[,2])

plot(xa.r,yb.r)

xa.t = rnorm(365)

yb.t.obs = rlnorm(365)

yb.t.sim = approx(x = sort(xa.r), y = sort(yb.r), xout = sort(xa.t))$y

plot(1-ppoints(365,0),sort(yb.t.obs) , type="l", log="y")

lines(1-ppoints(365,0), yb.t.sim ,col=2)

############################################
```

---

## Short Comment (SC2) · 20 Jul 2018

This is just an update of the code including correlation. In the previous version I forgot to correlate x and y in the target period. Sorry!

```
**with perturbation and some cross-correlation**

xa.r = rnorm(365*20)+rep(rnorm(20,sd=0.4),each=365)

yb.r = rlnorm(365*20)*rep(rlnorm(20,sd=0.6),each=365)

rho = 0.2

corM <- rbind(c(1.0, rho), c(rho, 1.0))
```

```
SigmaEV <- eigen(corM)

Meps <- rbind(xa.r,log(yb.r))

Meps <- SigmaEV$vectors %*% diag(sqrt(SigmaEV$values)) %*% Meps

Meps <- t(Meps)

**plot(Meps)**

xa.r = Meps[,1]

yb.r = exp(Meps[,2])

plot(xa.r,yb.r)

xa.t = rnorm(365)

#############

yb.t.obs = rlnorm(365)

Meps2 <- rbind(xa.t,log(yb.t.obs))

Meps2 <- SigmaEV$vectors %*% diag(sqrt(SigmaEV$values)) %*% Meps2

Meps2 <- t(Meps2)

**plot(Meps2)**

xa.t = Meps2[,1]

yb.t.obs = exp(Meps2[,2])

yb.t.sim = approx(x = sort(xa.r), y = sort(yb.r), xout = sort(xa.t))$y

plot(1-ppoints(365,0),sort(yb.t.obs) , type="l", log="y")

lines(1-ppoints(365,0), yb.t.sim ,col=2)
```

---

## Short Comment (SC3) · 21 Jul 2018

The comment was uploaded in the form of a supplement: https://www.hydrol-earth-syst-sci-discuss.net/hess-2018-347/hess-2018-347-SC3-supplement.pdf

---

## Short Comment (SC4) · 21 Jul 2018

Dear Francesco,

thank you for your fast and interesting comments. They are certainly useful, but I have the impression that you overlooked some parts of the paper, and for sure missed the essence of the idea. To make this clear I use a slightly more formal way. We assume that discharge is somehow related to precipitation. Formally

$$Q_k(t) = h_k(P_k(t - \tau), \tau = 0, \ldots, n \ \ldots, \ \beta_k)$$

where $k$ stands for the location, $h_k$ is the transformation - usually approximated by a

hydrological model - $\beta$ are the specific parameters of the process.

Our goal is to find the distribution of $Q_k(t)$ for a time period $(T_1, T_2)$ (FDC). We would like to have a quick approximation without hydrological modelling. (Modelling is unfortunately often introducing additional errors, and is often biased for longer subperiods. Thus it is complicated in its error structure.) Obviously we have a correlation between $P_k$ and $Q_k$. Unfortunately we cannot use the distribution of $P_k$ to assess the FDC directly as it will fail due to the lacking temporal structure and the many zeros. (Thus your statement any correlated series would do the same job is false.) Instead we use a transformation of $P_k$ - the API:

$$A(t) = a_k(P_k(t - \tau), \tau = 0, \ldots, n)$$

Both transformations can be regarded as filters acting on $P_k$. These filters do not necessarily produce highly correlated series, but may produce series with similar distributions. The trivial example for this is if we assume that

$$h_k = a_{k-m}$$

This is a time shift - typical for hydrological processes. The two series are not perfectly or weakly correlated, but their distributions are practically the same.

Thus the basic idea was to get rid of the complicated non-linear processes and to find a filter which relates the distributions.

Personally I am not very fond of the simple correlation type approaches

I fully disagree with your statement:

*To be honest, I think that whatever variable recorded in the reference and target period can be used as a support for interpolation, if its realizations in the reference period explore the sample space in the same way as the realizations of the target variable do.*

One important message of the paper is that FDCs can be very different from time period to time period. (The usual assumption that they and the related indices are characteristic for the catchment is not true.) Thus only parameters showing the same kind of signal may work. API was selected as it reflects the same generating mechanism.

You may have overlooked, but we tried the same procedure with FDCs from other sites too. Here some (most likely where $h_k$ and $h_l$ were similar) worked well others not.

Unfortunately I do not use, thus do not understand R-codes. Thus I cannot comment on your code. It would be interesting if you would modify or permute the precipitation series before assessing the performance of this very simple method.

Regards Andras

---

## Short Comment (SC5) · 21 Jul 2018

Dear Andras,

Thanks for your reply. Now my question is: why things you explained so clearly in few lines in your reply do not seem even mentioned in the paper? According to your response, the key point is that API shows a distribution close to stream flow (as it reflects the same generating mechanism). However, I cannot see this in the very short section 3.2, and figures, equations, etc. seem not to show this key fact, at least explicitly. On the other hand, there is a long discussion in section 3.3 and an appendix describing what is nothing but a quantile mapping, which should not be the main point of the paper, if I correctly interpret your response.

You are right when you say that my statements about independence are incorrect; in fact, in the second post that I sent, I showed diagrams (in the first figure) where I show the effect of correlation on the reproduction of 'FDCs'.

In my opinion, here the point is that the proxy variable you chose (whatever it is, apart some generic requirements) should closely follow the fluctuations of the target variable in order the quantile mapping to work, which is also what implies your first equation  $Q_k(t) = A(t)$ , or a less demanding  $Q_k(t) \propto A(t)$ . In this respect, I cannot see how your sentence 'The two series are not perfectly or weakly correlated, but their distributions are practically the same.' applies in this context.

In fact, what matters in a quantile mapping is, I think, that the variables co-move in order to guarantee quantile matching (Figure 2 of Smakthin and Masse (2000) provides a nice illustration), and this seems to me obvious. Quantile mapping in the appendix does not need that the distributions of Q and A are the same; instead, they can be arbitrary, apart from minimal general requirements (in the second figure of my previous comment I used normal and lognormal CDFs for X and Y, respectively). Quantile mapping works because low/middle/high quantiles of X correspond to (and tend to occur in your time sequence jointly with) low/middle/high quantiles of Y, irrespective of the shape of their marginal distributions (with usual general assumptions on continuity, etc.). Ideally, the matching is perfect if the variables are comonotonic: the weaker the co-movement (correlation) the worse the effectiveness of quantile mapping (figure below attempts to illustrate what I mean). If Q and A are not well correlated, even if the marginals of Q and A were identical, there would be no way to know which quantiles of Q occurred in a target period from the observed quantiles of A in the same period, because the latter can be e.g. systematically low/high for some reason, while the former can be whatever, and vice versa. In this respect, I think that the quality of your results depends on how closely A co-moves (is correlated) with Q, and nothing else. So, I repeat: every variable well correlated with Q does the job (of course, A can be one of them).  $P_k$  does not because the correlation with Q is poor, and  $P_k$  is discrete-
continuous while Q is generally continuous (...leaving aside intermittent streams, etc., you know). Thus, in my comment, it was implicit that the proxy should share minimal properties with Q in order to closely follow flow fluctuations.

Thus, if the message is that a weighted moving average of antecedent rainfall (API) closely co-moves with streamflow and thus A is a very good proxy enabling reasonably reliable quantile matching, that's fine. However, in this case I think that the paper should be reworded to focus on this, by preliminary showing the evidence of this assumption (as also suggested by Dr. Farmer). In the present form, as you can see by yourself, most of the test is devoted to (i) description of KS tests adjusted in a rather rough manner and applied in a context where every underlying assumption is not fulfilled, (ii) quantile mapping, which is not even performed in a consistent way, and in turn can be summarized by a single equation,  $q = \hat{F}_Q^{-1}(\hat{F}_A(a))$ , strongly shortening and simplifying (if not removing) section 3.3, and (iii) a long list of well-known performance indices whose selection looks a bit random, at least to me. On the other hand, the only equation of interest and few explanatory sentences reported in your reply are not shown... I may have missed them.

To summarize, if you think that the co-movement of *Q* and *A* is new and important (along with the message that FDCs depends on both meteorological forcing and basin response... is this not known?... Anyway), and represents something to be communicated, that's fine. As you know, as I'm statistically biased, for me *A* is just a good covariate/proxy, even though I recognize that it is chosen looking at the generating mechanism, and can be of practical interest for HESS audience. However, this should be stated much better in the text. Nonetheless, please forgive me if I cannot see what is interesting and new in the roughly performed and badly described quantile mapping, KS testing for ni/nid variables, apparently random selected measures of performance, and if I find the overall presentation very poor in terms of material organization, mathematical notation, clarity, and synthesis. I think that the empirical considerations/arguments (about FDC fluctuations, etc.) and statistical techniques can be compactly and clearly
summarized in a few pages using simple and effective mathematical notation (as you are a mathematician, you know what I mean).

Honestly, I am perplexed. This is not your standard, and I am confident that you are conscious of this, even if you will legitimately defend this work. By my side, I am not a reviewer; so, my opinion does not matter very much. I am confident that you do not mind if expressed it with my usual rough tone.

As ever

F

Dear Andras,

Thanks for your reply. Now my question is: why things you explained so clearly in few lines in your reply do not seem even mentioned in the paper? According to your response, the key point is that API shows a distribution close to stream flow (as it reflects the same generating mechanism). However, I cannot see this in the very short section 3.2, and figures, equations, etc. seem not to show this key fact, at least explicitly. On the other hand, there is a long discussion in section 3.3 and an appendix describing what is nothing but a quantile mapping, which should not be the main point of the paper, if I correctly interpret your response.

You are right when you say that my statements about independence are incorrect; in fact, in the second post that I sent, I showed diagrams (in the first figure) where I show the effect of correlation on the reproduction of 'FDCs'.

In my opinion, here the point is that the proxy variable you chose (whatever it is, apart some generic requirements) should closely follow the fluctuations of the target variable in order the quantile mapping to work, which is also what implies your first equation  $Q_k(t) = A(t)$ , or a less demanding  $Q_k(t) \propto A(t)$ . In this respect, I cannot see how your sentence 'The two series are not perfectly or weakly correlated, but their distributions are practically the same.' applies in this context.

HESSD
In fact, what matters in a quantile mapping is, I think, that the variables co-move in order to guarantee guantile matching (Figure 2 of Smakthin and Masse (2000) provides a nice illustration), and this seems to me obvious. Quantile mapping in the appendix does not need that the distributions of Q and A are the same; instead, they can be arbitrary, apart from minimal general requirements (in the second figure of my previous comment I used normal and lognormal CDFs for X and Y, respectively). Quantile mapping works because low/middle/high quantiles of X correspond to (and tend to occur in your time sequence jointly with) low/middle/high quantiles of Y, irrespective of the shape of their marginal distributions (with usual general assumptions on continuity, etc.). Ideally, the matching is perfect if the variables are comonotonic: the weaker the co-movement (correlation) the worse the effectiveness of quantile mapping (figure below attempts to illustrate what I mean). If Q and A are not well correlated, even if the marginals of Q and A were identical, there would be no way to know which quantiles of Q occurred in a target period from the observed quantiles of A in the same period. because the latter can be e.g. systematically low/high for some reason, while the former can be whatever, and vice versa. In this respect, I think that the quality of your results depends on how closely A co-moves (is correlated) with Q, and nothing else. So, I repeat: every variable well correlated with Q does the job (of course, A can be one of them).  $P_k$  does not because the correlation with Q is poor, and  $P_k$  is discretecontinuous while Q is generally continuous (...leaving aside intermittent streams, etc., you know). Thus, in my comment, it was implicit that the proxy should share minimal properties with Q in order to closely follow flow fluctuations.

Thus, if the message is that a weighted moving average of antecedent rainfall (API) closely co-moves with streamflow and thus *A* is a very good proxy enabling reasonably reliable quantile matching, that's fine. However, in this case I think that the paper should be reworded to focus on this, by preliminary showing the evidence of this assumption (as also suggested by Dr. Farmer). In the present form, as you can see by yourself, most of the test is devoted to (i) description of KS tests adjusted in a rather rough manner and applied in a context where every underlying assumption is not fulfilled,
(ii) quantile mapping, which is not even performed in a consistent way, and in turn can be summarized by a single equation,  $q = \hat{F}_Q^{-1}(\hat{F}_A(a))$ , strongly shortening and simplifying (if not removing) section 3.3, and (iii) a long list of well-known performance indices whose selection looks a bit random, at least to me. On the other hand, the only equation of interest and few explanatory sentences reported in your reply are not shown... I may have missed them.

To summarize, if you think that the co-movement of Q and A is new and important (along with the message that FDCs depends on both meteorological forcing and basin response... is this not known?... Anyway), and represents something to be communicated, that's fine. As you know, as I'm statistically biased, for me A is just a good covariate/proxy, even though I recognize that it is chosen looking at the generating mechanism, and can be of practical interest for HESS audience. However, this should be stated much better in the text. Nonetheless, please forgive me if I cannot see what is interesting and new in the roughly performed and badly described quantile mapping, KS testing for ni/nid variables, apparently random selected measures of performance, and if I find the overall presentation very poor in terms of material organization, mathematical notation, clarity, and synthesis. I think that the empirical considerations/arguments (about FDC fluctuations, etc.) and statistical techniques can be compactly and clearly summarized in a few pages using simple and effective mathematical notation (as you are a mathematician, you know what I mean).

Honestly, I am perplexed. This is not your quality level, and I am confident that you are conscious of this, even if you will legitimately defend this work. By my side, I am not a reviewer; so, my opinion does not matter very much. I am confident that you do not mind if expressed it with my usual rough tone.

As ever

F

PS: API patterns look like simple and empirical counterparts of shot noise processes

**HESSD**
used to model stream flow data

F. Laio, A. Porporato, L. Ridolfi, and I. Rodriguez-Iturbe (2001) Mean first passage times of processes driven by white shot noise Phys. Rev. E 63, 036105

Claps, P., A. Giordano, and F. Laio (2005), Advances in shot noise modeling of daily streamflows, Adv. Water Resour., 28, 992–1000.

**HESSD**
Fig. 1.

---

## Referee Comment (RC2) · T. Over (Referee) · 4 Sep 2018

Overview:

This paper presents a method for estimating FDCs during an ungauged period at a "target" location that is gauged during another period. It is based on the antecedent precipitation index (API) at another ("reference") basin, where a correspondence is set up between the FDC at the target gauge during its gauged period and the FDC of the API at the target basin and between the FDCs of the API at the reference basin during the gauged and ungauged periods. As such it is similar to a record extension application of the approach of Smakhtin and Masse (2000, Hydrological Processes, Vol. 14, pp. 1083-1100), except they estimated daily flow in ungauged basins, and to the work

of Hughes and Smakhtin (1996, Hydrological Sciences Journal, vol. 41, pp. 851-871) who presented the use of time series of discharges at target and reference basins and their FDCs for daily record extension. Combining those two contributions yields the use of CPI or API for record extension, as was used by Straub and Over (2010) in an admittedly obscure report at: https://apps.ict.illinois.edu/projects/getfile.asp?id=3033, appendix A, section 3.2, and possibly elsewhere.

Arguably, while it seems the methodology in this paper could be considered a corollary of the prior work cited above, the focus of the present on FDCs rather than daily stream-flow may distinguish it. Ideally, however, the authors would investigate the distinction and show how in application one might make different choices regarding parameters or selection of the reference basin when estimating the FDC as opposed to daily stream-flow.

Apart from that overlap with previous work, I have the following major concerns regarding this paper in its current form: 1. As mentioned, in the methodology presented, the API at another gauged basin, the "reference" basin, is used for the FDC extension at the target basin during its ungauged period. This is strange, for two reasons: (1) Why select another gauged basin and use only its API, not its streamflow, and (2) Why not use the API at the target basin? I am in fact rather surprised it seems to work as well as it does when the API used is not at the target basin. Perhaps this points to a difference between the application for daily streamflow as compared to FDCs (i.e., that the FDC application is comparatively forgiving).

2. A list of detailed comments is given below, but in general the presentation is rather uneven; some of the primary issues are:

a. The idea of using the reference gauge discharge is raised and in the first presentation of the methodology on page 10, the FDC at the reference basin is computed, but it is never used.

b. The results for the US basins are presented quite differently than those of the Ger-

**HESSD**
man basins, including using different performance criteria.

c. How the performance criteria could be applied for individual predictions was not clear to me.

d. The choice of basins for the study seems rather arbitrary. For example, there are hundreds of basins in the MOPEX, including many others that do not have much snow.

e. The consideration of energy versus water limitation as a measure of similarity is interesting but it is not clear that it is relevant when API is being used.

Specific technical comments:

1. Section 2.1:

a. Are there snow effects in the Upper Neckar basin? How addressed?

b. Do you take the karstic effects on Upper Neckar flows into account?

- 2. Section 2.3
- a. Figure 3:

i. Perhaps plot and check correlation with temperature of ET/P instead of just ET?

ii. Need to consider uncertainty around correlation estimates: for the Peace R. it seems unlikely that rho = 0.027 is a significantly positive value; rather probably this basin is balanced between energy and water limitation by this criterion.

b. Last sentence on p. 6: It is not possible ...." It sounds plausible, but has this assertion been tested? The statement itself is very categorical; in fact there are degrees of water and energy limitation. How different do they need to be to make this true (if it is)? In particular, for the present application, the methodology may account for the water versus energy limitation; it may be that the timing of the weather is the most important thing to have in common.

3. Section 3.1
a. 2nd paragraph:

i. It seems there should already be a well-established way of addressing autocorrelation effects on the K-S test.

ii. The last two sentences of this paragraph seem to be referring to a test on a particular basin, but they are stated as if these relations are generally (i.e., mathematically) true. Which is it?

b. 3rd paragraph: This paragraph seems to include "Results", not "Methodology".

4. Section 3.2:

a. First paragraph: It is not always true that the non-weather properties (land use) do not change. Did you check that your study basins satisfy this assumption?

b. Last sentence: Did you test different values of alpha other than 0.85, or just select that value for the reasons given?

5. Section 3.3:

a. First paragraph, last sentence: Why do you assume "large scale precipitation"? What do you mean by that?

b. Last complete paragraph on p. 10: It seems it would be better to interpolate between Pj and Pj+1 rather than taking the mean, but it may not make a lot of difference.

6. Section 4, p. 11, discussion of figures 5-8:

a. Several statements regarding goodness of fit are made without being quantified. However the K-S technique has been presented and could be applied; indeed, it would be ideal to provide K-S test results to accompany the results in each panel of these plots.

7. Section 4, p. 13, figure 10 and discussion of it:

a. Why present 30, 70, 90, and 99th percentiles? As one can see, 90th and 99th
(though the lower right panel of figure 10 is labeled as the 95th percentile), are almost the same. The complementary percentiles, i.e., 70, 30, 10, and 1st percentiles (exceedance probabilities) would be more interesting, in my opinion.

b. You say (lines 5-6 of p. 13): "it is not possible to estimate the flow quantiles using regression methods that do not take into account the weather characteristics." This may be an over-statement. You have demonstrated that if you want to transfer across time, weather fluctuations need to be considered. But for prediction at ungauged basins for a fixed period of time, that may not be true.

Comments on the presentation

1. Section 3.1, 3rd paragraph: This paragraph seems to include "Results", not "Methodology".

2. Section 3.3, in steps 2&3 of the "procedure to predict" (p. 10), the FDCs of the reference catchment A is computed, but it does not seem to be used in the procedure.

3. Section 3.3, step 8 of the "procedure to predict" (p. 10): Suggest "qBrj is taken to be the value of discharge that occurred..." rather than simply "qBrj is the value of discharge that occurred..."

4. Section 3.3, last paragraph (p. 11): It is stated here that in the paper both discharge and precipitation will be used as the support variable. But everything before indicates that only precipitation will be used. And I don't see any results using discharge as the support variable.

5. Section 4, figures 5-8:

a. From what period is this FDCref\_site that is plotted? As it does not seem to be used in the procedure, why is it plotted?

b. I think however you should add the FDC of the target site during the reference period to these plots so the reader can see how much the FDC has changed from reference
period to target period.

6. Section 4, figure 9 and discussion of it:

a. Discussion of figure 9 on p. 12, lines 9&10: "Results shown that the distance between the former pairs is bigger than the distance between the latter, Figure 9.":

i. I don't think you ever defined the K-S distance. That needs to be done.

ii. I am willing to believe this assertion is true, but it is hard to see just from the plot. Can you provide some summary results such as the mean and median difference between 9 (top) and 9 (bottom) to give evidence of the assertion.

iii. This assertion should be restated without the shorthand of "former" and "latter". It is hard to understand the way it is currently phrased, and it is a very important point.

7. Section 4, pp. 11-14: It is not clear why the Results section starts by giving a lot of results for the U.S. catchments and none for the German ones.

8. Section 4.1, pp. 14-17, Definition of performance criteria:

a. Are all these computed for Q in mm units? Even though those are units used throughout, it would be worth re-emphasizing that here.

b. BIAS: This is not a simple bias as it is normalized by Qsim; it is more like a relative bias or "relative mean error"; however usually one divides by Qobs. Actually, ME (defined later) is more like a simple bias.

c. Why apply different criteria for the German catchments?

d. "Ratio":

i. Can you give it a more meaningful name?

ii. This formula looks odd. If the square root were only on the numerator, it would be the standard error divided by the mean error (and the quantity would be non-dimensional). But why apply the square root to the mean error in the denominator?
9. Section 4.1, figures 11-13: Many of the colors these figures are shifted so each box has more than one color, making them hard to interpret. This effect needs to be fixed.

10. Section 4.1: I don't see how the Performance criteria were applied to create the results shown in figures 11-13. As I understand, there is only one prediction of each quantile, for a fixed reference catchment and decade. Then how does one do the summations indicated in the performance criteria formulae? Following the definition of NSE on p. 15 it says: "N is the number of discharge values related to a specific percentile". How many of those are there? Is there ever more than one? If so, how? The situation with the correlation coefficient values presented in figure 14 seems to be the same: How does one compute correlation coefficients without multiple values? If there are multiple values, where are they coming from?

11. Conclusions:

a. p. 21, lines 2-5: "Here it is shown that two FDCs built for the same catchment, but with data corresponding to two different time windows, cannot be regarded as the same continuous distribution. The same results when two FDCs of two different catchments built for the same time window are analysed. Thus, it is not possible to infer a FDC using parameters retrieved from the distribution of another FDC without considering the weather." The first sentence supports the assertion in the third, but the second does not. If two different catchments experience possibly similar weather but produce a different streamflow, the cause is not the weather.

b. p. 21, lines 13-14: "Since precipitation data series are characterized by a high number of zeros, here we used the Antecedent Precipitation Index (API)." This statement misses the more important fact that the API combines in a streamflow-like way the history of the precipitation. (A similar statement is made at the beginning of section 3.2 near the bottom of p. 8.)

c. p. 22, lines 26-27: Qualitative statement about similarity in shape from beginning of Section 4 is repeated. This assertion needs to be quantified somehow.

HESSD

---

## Author Comment (AC1) · 29 Jul 2019

Reply to Reviewer 1

We would like to thank William Farmer for the time dedicated to our paper and for his comments that will contribute to improve the paper.

In the following, reviewer's comments are in Italic (R1), Authors' comments are in normal text (AC). Moreover , authors' changes in manuscript based on comments of all referees are summarized at the end of this document.

*R1: This work could benefit from a clear statement of hypotheses. In my opinion, the main hypothesis is that the daily flow duration curves at an ungauged location can be simulated with knowledge of the precipitation record at both the ungauged site and some index site. This hypothesis relies on a further assumption that the cumulative distributions of streamflow and precipitation correlate in some way. In the revisions I am proposing, I think the authors should clearly set out to quantifiably address these hypotheses. A figure may improve the understanding of the methodology.*

AC: We thank the reviewer for the suggestion, We thank the reviewer for the suggestion, we will clearly state the hypothesis of the work (see also comments below) and a figure will be added to give a clearer explanation of the procedure.

*R1: As a previous commenter noted, the approach is difficult to understand and may be greatly simplified. The authors create cumulative distribution functions (CDFs) for streamflow and API at both an index site and a target site over some reference period. They then create a CDF of streamflow and API at the index site for some target period and a CDF for API at the target site for this same target period. The method then only uses the CDFs of (1) API at the index site in the target period, (2) API at the index site in the reference period, and (3) streamflow at the target site in the reference period.*

AC: We thank the reviewer for the comment, we will change Sect. 3 to make it clearer and shorter and we will remove redundant parts.

*R1: In addition to improving readability, revising the methods section might also address the concerns raised by the previous commenter. Namely, that it seems the approach could be greatly simplified through interpolation along the relevant relationships without the need for intermediate exceedance probabilities. This would be accomplished by the following: (1) Create the CDF of API at the index site in the target period, (2) Plot the API of the index site in the reference period against the streamflow at the target site during the reference period, (3) interpolate each API from the target period, in order, along the curve created in (2) to produce the CDF of streamflow at the target site in the target period. While this approach is achievable algorithmically, and identical to the one proposed, it raises several concerns about the implicit assumptions. "Are API and streamflow ranked independently? (See step 3 of page 10.) If so, the implicit assumption is that the exceedance probability of the API on a given day is equivalent to the exceedance probability of the streamflow on that same day at the same site. This is a pretty sizeable assumption. As a start, it would be good*

*to see if the temporal sequence of API exceedance probabilities is highly correlated with the temporal sequence of streamflow exceedance probabilities at a single site over the same period."*

AC: We thank the reviewer for the comment, the methodology section will be largely modified and a clear statement of the hypothesis will be provided. Specifically, it will be shown that the temporal sequence of API exceedance probabilities is highly correlated with the temporal sequence of streamflow exceedance probabilities at a single site over the same period. For instance, in the following a table showing this correlation is reported for Blanco River (USA) for four different periods of time.

| Period | correlation |
|---|---|
| 1948-1968 | 0.978 |
| 1968-1988 | 0.995 |
| 1948-1963 | 0.998 |
| 1948-1958 | 0.970 |

*R1: The second assumption arises when we move from the CDF of API at the index site in the reference period to the CDF of streamflow at the target site in the reference period. This movement introduces a second implicit assumption: namely, that the exceedance probability of the API on a given day at the index site is equivalent to the exceedance probability of streamflow on that same day at the target site. Put another way, if you accept the assumption in the previous paragraph, this step assumes that the temporal sequencing of API is identical at both sites. Again, this needs to be demonstrated: Is the temporal sequence of API exceedance probabilities at the index site highly correlated with the temporal sequence of streamflow exceedance probabilities at the target site in the reference period? It may be argued that the temporal sequencing is irrelevant. This is not the case. By assuming the same exceedance probabilities in step 7 of page 10, we are assuming a perfect correlation and, therefore, assuming a temporal correspondence.*

AC: We thank the reviewer for the comment, in the revised manuscript we will show that the temporal sequence of API exceedance probabilities at the index site is highly correlated with the temporal sequence of streamflow exceedance probabilities at the target site in the reference period. As an example, in the following a table reports the correlation between Blanco (index site) and three target sites (specified below) for four different reference periods.

| Sites | 1948-1968 | 1968-1988 | 1948-1963 | 1948-1958 |
|---|---|---|---|---|
| Tangipahoa | 0.996 | 0.997 | 0.994 | 0.997 |
| Choctwhatchee | 1 | 1 | 0.999 | 0.999 |
| Bogue | 0.990 | 0.992 | 0.995 | 1 |

*R1: The third assumption, which was alluded to earlier, is that the CDF of API is identical across sites for both the index site and the target site in the same period. This is what allows the authors to use the CDF of the API of the index site in the target period for step 6 on page 10. It may be that this is what the authors meant by "the assumption of large scale precipitation" (line 16, page 9); if so, please clarify. Regardless, this assumption needs to be validated through correlation or a KS test.*

AC: We thank the reviewer for the comment, in the revised manuscript we will show that the CDF of API is identical across sites for both the index site and the target site in the same period through correlation. In the table below an example of correlation between Blanco (USA) and three other sites is reported for four different time periods.

| Sites | 1948-1968 | 1968-1988 | 1948-1963 | 1948-1958 |
|---|---|---|---|---|
| Tangipahoa | 0.98 | 0.99 | 0.97 | 0.99 |
| Choctwhatchee | 0.98 | 0.98 | 0.98 | 0.98 |
| Bogue | 0.99 | 0.99 | 0.98 | 0.99 |

The distributions of API have the same type of distribution – Weibull is accepted for all of them. On the other hand the distribution parameters may differ from site to site and from time period to time period.   For instance, the Weibull is the best fitting distribution of the API at Blanco for the periods 1948-1968, 1968-1988, 1948-1963 and 1948-1958. The same applies to the API of three sites for the same periods as it is shown from the table below.

| Sites | 1948-1968 | 1968-1988 | 1948-1963 | 1948-1958 |
|---|---|---|---|---|
| Tangipahoa | Weibull | Weibull | Weibull | Weibull |
| Choctwhatchee | Weibull | Weibull | Weibull | Weibull |
| Bogue | Weibull | Weibull | Weibull | Weibull |

*R1: Without some quantifiable validation of these assumptions, the proposed method is tenuous at best and left vulnerable to criticism. With that in mind, and the comments of the previous commenter, I'd like to propose that exploring these assumptions might result in modifications of the methodology that might move away from the case of simple interpolation. Is the relationship between API and streamflow constant across periods or sites? Should API and streamflow be ranked independently or with some sort of dependence? Should the API of the index site in the target period be used to map to a different site in a different period (i.e., the target site in the reference period)? Exploring these questions, and validating the underlying assumptions, will produce a more robust approach."*

AC: Thank you for this comment. We assume that the relationship between API and FDC is the same for the same site regardless the time period. This assumption has to be justified, and we'll add a paragraph on this issue to the revised manuscript.

*R1: "In addition to their main hypotheses proposing this methodology, the authors assert that the FDC is a product of the basin and the weather. This is surely intuitive, but the evidence provided could be greatly strengthened. The authors use KS tests, but is unclear how they were applied. It would be informative to clearly communicate if the CDF of streamflow from one period and the CDF of streamflow from another period could be considered significantly different. The authors have done this, but the presentation is not clear. The extension would be to ask if the API can be correlated with any differences across time. (As an aside: Was there any discussion of selecting stationary sites? How would nonstationary behavior play a role here?)"*

AC: The FDC seems unfortunately to be significantly different from one time period to the other. The same applies to the API. In our opinion this is not caused by non stationarity of the time series but

more to some long memory effects. This of course has to be checked more in detail and a corresponding discussion will be added to the paper.

*R1: This, in my opinion, raises another concern: The authors seem to be attempting to simultaneously address two very different problems. The first problem considers a target site that has a streamflow record overlapping with an index site, but the desired period has no overlap (the ungauged area is the same site, different period). In this case, the use of APIs within site, without an index site, would be most ideal. The second problem considers a site without any streamflow information; this situation necessitates the use of an index. Of course, when there are gaps in the API record as well, this transforms into four unique problems. Regardless, if we believe the underlying assumption that the CDF of streamflow is a product of basin and weather, then the solutions to these problems must be quite different. The first asks if knowledge of new weather can produce the CDF of streamflow, while the second alters both variables and asks if the CDF relationship can be transferred across weather and basin. Line 8 of page 3 implies that both problems are considered, but the remainder of the paper seems only to address the partially gauged site. I would advise addition of the second problem or, at least, a discussion of implication for the second problem (completely ungauged).*

AC: We thank the reviewer for the comment, the methodology explained in the paper needs to be applied to partially gauged basins, we will eliminate the reference to totally ungauged sites in the paper.

*R1: In 1996, Hughes and Smakhtin (<https://doi.org/10.1080/02626669609491555>), among others, provided a technique for hydrograph simulation using flow duration curves. While their focus was on hydrographs, the extensions to ungauged FDCs can be made quite clearly (i.e., they could be derived from simulated hydrographs). Smakhtin and Masse (2000: <https://doi.org/10.1002/(SICI)1099-1085(20000430)14:6%3C1083::AID-HYP998%3E3.0.CO;2-2>) then extended this method to use a precipitation index. While I believe that the methods presented here are different, the novelty of this new method must be strongly articulated.*

AC: The two papers recalled by the reviewer are very interesting and we will recall them in the Introduction section, strongly stating which is the novelty behind our work. We will add the following sentences in the manuscript:

"Smakhtin and Masse [1] used the weather at a donor site to extend the daily hydrograph at a destination site through the monthly FDC of the destination site itself. The monthly FDC at the "destination" site is found using different methods such as (i) regionalization of FDCs based on available observed records from several adjacent gauges Smakhtin et al. [2] or (ii) conversion of FDCs calculated from monthly data into 1-day FDCs (Smakhtin, [3]). The procedure presented by Smakhtin and Masse [1] is an extension of a previous work proposed by Hughes and Smakhtin [4] to extend and/or filling in daily flow time series. The drawback of the procedures proposed by Hughes and Smakhtin [4] and Smakhtin and Masse [1] is the necessity of retrieving the monthly FDC of the target site with well-known literature methods before applying the methodology to extend the hydrograph. While the novelty of the approach we propose is the possibility to retrieve the FDCs at partially ungauged sites from weather recorded at a reference catchment. Thus, without the need of applying procedures such as the regionalization."

R1: *"I strongly encourage the authors to revisit the style of the manuscript. At times, it feels a bit disjointed and it may be improved by enforcing a strict Introduction-Methods-Results-and-Discussion (IMRAD) format. For example, section 3.1 is ostensibly a methods sections but presents a series of results that I think are pivotal to the paper (line 19, page 8). Similarly, the paragraph on page 13 and section 4.1 present new methods of analysis that have not been presented earlier in the paper. While IMRAD is not a requirement, I do suggest thinking carefully about the best approach to presenting the narrative."*

AC: We thank the reviewer for the suggestion, a restyle of the paper will be performed. The results reported in Sect.3.1. are pivotal to the development of the paper, they will be moved under a paragraph named "Preliminary analysis".

The presentation of the performance criteria will be moved into the Methodology

R1: *"In my opinion, this work needs more presentation and discussion of quantified results.*

AC: In the updated version of the paper we will further develop the discussion.

R1: *"The results sections heavily rely on visualization. Even the presentation of metrics in section 4.1 is visual. While this is useful, we still need to see some discussion on the performance metrics. For example, the scale on NSE in Figure 11 makes all positive values appear as a single color. This presentation means we can't honestly see how the methods perform."*

AC: We chose to present results in a visual style to better show them. Because of the large number of sites and the large number of time windows we have investigated, to a reader it would take too much time to go through a tabular presentation, which may be also boring, while plots have an immediate impact. However, we agree with the reviewer regarding the scale of the plots, that sometimes make difficult to understand the goodness of the results. We will solve this issue in the paper, moreover we will add a deeper discussion of the results.

R1: *Page 1, line 17: When talking about general duration curves, more commonly known as cumulative distribution functions, it is better to say "exceedance frequency" rather than "exceedance time".*

AC: We will replace the word as suggested.

R1: *Page 2, line 1: Please provide the citation for the Weibull plotting position.*

AC: The citation will be added (i.e., Weibull, W., 1939: A statistical theory of the strength of materials. Ing. Vetensk. Akad. Handl., 151, 1–45).

R1: *Page 3, line 4: Please provide more discussion and literature of this important point.*

AC: The sentence refers to the results of the paper anticipating them for the readers. A deeper discussion is provided in Sect. 3.1. We will made it clearer in the text: "Results show that the FDC is not a property of a specific basin, because different time windows (such as decades) relate to different FDCs."

*R1: Page 3, line 7: It is not clear what the "distribution of the FDC" is. The FDC is a distribution, so it is confusing to talking about the distribution of a distribution.*

AC: We rephrased the sentence as "It is not possible to develop relations between basin parameters and FDC characteristics to yield synthesized FDCs where flow data are not available, as done for instance by Quimpo et al. [5]."

*R1: Page 4, line 6: Florida, Louisiana and Texas are certainly not the East coast. I would suggest the Gulf Coast.*

AC: We will rephrase the sentence

*R1: Page 5, line 7: misspelling of database*

AC: We will carefully check the English spelling throughout the paper.

*R1: Page 5, line 2 (?): This is an example of inconsistent citation style. Bloeschl should be in parenthesis.*

AC: We will carefully check the citation style throughout the paper.

*R1: Page 8, line 5: Please provide citation to KS test.*

AC: We will provide the citation (i.e. Massey, F. J. The Kolmogorov-Smirnov Test for Goodness of Fit. Journal of the American Statistical Association. Vol. 46, No. 253, 1951, pp. 68–78.).

*R1: Page 9, line 10: What lead to this choice for alpha? (Also, note that the same symbol is used earlier in this section for significance: page 8, line 10.) Please provide citation or summary of initial exploration.*

AC: We will provide different symbols for the two. When $\alpha$ tends to zero, API keeps tracks of the precipitation occurred in the few previous days and it represents the short memory of the basin. When $\alpha$ tends to 1, API represents the long memory of the basin as it includes the effect of precipitation occurred many days before. To capture this behavior, in this study $\alpha$ is chosen equal to 0.85. Moreover, this is in agreement with a previous study [6] which investigated the same case study area (i.e. Neckar catchment). This will be specified in the paper.

*R1: Page 9, line 18: I strongly suggest referring to the "reference site" as an "index" or "donor". The reference connotation implies lack of human influence that might be confusing. The same could be said of the reference period*

AC: We will use "donor" site instead of "reference" one.

*R1: Page 10, line 17: The series of Nr and Nt are both being indexed with i, which leads to confusion.*

AC: Since the reference series of both basins A and B have the same size, we will index the reference series (Nr) with $i$. The same applies to target series of both A and B, therefore we indexed the target series (Nt) with $j$. Moreover, we will consistently revise the Methodology section.

*R1: Page 10, line 19: So, API_Ati is equal to API_Arj?*

AC: Yes, it is. We will add the following Figure to better clarify the methodology and also we will re-organize and strongly improve the methodology section.

[Figure]

*R1: Page 11, line 4: What is a supporting variable? This is not described as such earlier, so it surprises the reader.*

AC: The supporting variable is the one used to retrieve the streamflow values, in this example is the API. We will better introduce it to be clearer.

*R1: Page 11, line 17: "good agreement" is very subjective. Please provide thorough, quantifiable analysis. For example, a lot of the curves in figure 6 look rather poor for highs and lows (top row, second box from the left).*

AC: At lines 17 and 18, the sentence is an introduction to the extensive explanation presented in the following lines of the manuscript. In the following, we comment each Figure, explaining weaknesses and strengths of each one. Thus, specifically, we comment the effect of choosing a small or large time period as reference period. Regarding Figure 6, we explicitly say that "The higher agreement in shape than in value is more evident when a small data set is considered as target period. For instance it results when the target period is 1 year, Figure 6. It is interesting to note that, even if for some target periods the simulated FDC does not perfectly match the observed one, the two FDCs have the same shape.

*R1: Figure 5: Why was the box for ref:68-88 and tar: 88-98 not included? The caption needs to do a better job of describing the different panels.*

AC: The missing panel would not add more information to the paper, therefore because of the lack of space the panel was not included. We will better describe the panels in the caption.

*R1: Page 12, line 9: Spelling of FDCs*

AC: We will carefully check the spelling throughout the paper.

*R1: Page 13: The methods for this paragraph were very unclear to me. Could a figure or a revision help?*

AC: In this paragraph we compute the moving average to show that the between-year variability of the discharge of a specific percentile can be high. Therefore, this suggests that percentiles cannot be considered an invariant characteristic of the basin and thus they cannot be estimated using geographic and morphologic characteristic of the basin only. We decided to show the moving averages of these percentiles as they are the most used ones.

*R1: Page 14: Please provide the citation for NSE. Even better, a metric like KGE might be more appropriate*

AC: The citation will be added. We used the NSE, despite the criticism it has received (e.g.,[7]) because of the familiarity most hydrologists and meteorologists have with it [8], facilitating the interpretation of the obtained values.

*R1: Figure 10: What is the horizontal axis of this figure?*

AC: Figure 10 will be replaced with a revised figure

**Author's changes in manuscript**

Some hints regarding authors' changes in the manuscript have been already given in the comments section. In the following, a summary of authors' changes in manuscript based on comments of all referees is given:

1. In the Introduction, we will clarify the novelty and the contribution of the paper.
2. The Methodology section will be reorganized and improved to provide a clearer description of the method. For the sake of clarity, a figure will be added.
3. We will provide a clear statement of the hypothesis.
4. The sections will be organized on the base of the Introduction-Methods-Results-and-Discussion (IMRAD) format.
5. For both case studies we will use the same performance criteria and results will be discussed in-depth.

*References*

[1]    V.Y. Smakhtin, B. Masse, Continuous daily hydrograph simulation using duration curves of a precipitation index, Hydrol. Process. 14 (2000) 1083–1100. doi:10.1002/(SICI)1099-1085(20000430)14:6<1083::AID-HYP998>3.0.CO;2-2.

[2]    V.Y. Smakhtin, D.A. Hughes, E. Creuse-Naudin, Regionalization of daily flow characteristics in part of the Eastern Cape, South Africa, Hydrol. Sci. J. 42 (1997).

[3]    V.Y. Smakhtin, Generation of natural daily flow time-series in regulated rivers using a non-linear spatial interpolation technique, Regul. Rivers Res. Manag. 15 (1999) 311–323.

[4]    D.A. Hughes, V. Smakhtin, Daily flow time series patching or extension: a spatial interpolation approach based on flow duration curves, Hydrol. Sci. J. 41 (1996) 851–871. doi:10.1080/02626669609491555.

[5]    R.G. Quimpo, A.A. Alejandrino, T.A. McNally, Region- alized flow duration for Philippines, J. Water Resour. Plan. Manag. 109 (1983) 320–330.

[6]     T. Sugimoto, Copula based Stochastic Analysis of Discharge Time Series, 2014.

[7]     H. V. Gupta, H. Kling, K.K. Yilmaz, G.F. Martinez, Decomposition of the mean squared error and NSE performance criteria: Implications for improving hydrological modelling, J. Hydrol. 370 (2009) 80–91.

[8]     D.N. Moriasi, J.G. Arnold, M.W. Van Liew, R.L. Bingner, R.D. Harmel, T.L. Veith, Model evaluation guidelines for systematic quantification of accuracy in watershed, Simulations, Trans. Am. Soc. Agric. Biol. Eng. 50 (2007) 885–900.

---

## Author Comment (AC2) · 29 Jul 2019

Reply to Reviewer 2

We would like to thank Thomas Over for the time dedicated to our paper and for his comments that contributed to improve the paper.

In the following, reviewer's comments are in Italic (R2), Authors' comments are in normal text (AC). Moreover, authors' changes in manuscript based on comments of all referees are summarized at the end of this document.

*R2: This paper presents a method for estimating FDCs during an ungauged period at a "target" location that is gauged during another period. […].As such it is similar to a record extension application of the approach of Smakhtin and Masse (2000, Hydrological Processes, Vol. 14, pp. 1083-1100), except they estimated daily flow in ungauged basins, and to the work of Hughes and Smakhtin […] Ideally, however, the authors would investigate the distinction and show how in application one might make different choices regarding parameters or selection of the reference basin when estimating the FDC as opposed to daily streamflow.*

Regarding the two papers recalled by the reviewer, Smakhtin and Masse [1] used the weather at a donor site, represented by the current precipitation index (CPI), to extend the daily hydrograph at a destination site through the monthly FDC of the destination site itself. The monthly FDC at the "destination" site is found using different methods such as (i) regionalization of FDCs based on available observed records from several adjacent gauges Smakhtin et al. [2] or (ii) conversion of FDCs calculated from monthly data into 1-day FDCs (Smakhtin, [3]). The procedure presented by Smakhtin and Masse [1] is an extension of a previous work proposed by Hughes and Smakhtin [4] to extend and/or filling in daily flow time series. The drawback of the procedures proposed by Hughes and Smakhtin [4] and Smakhtin and Masse [1] is the necessity of retrieving the monthly FDC of the target site with well-known literature methods before applying the methodology to extend the hydrograph. While the novelty of the approach we propose is the possibility to retrieve the FDCs at partially ungauged sites from weather recorded at a reference catchment. Thus, without the need of literature procedures such as the regionalization. We will recall the two papers by Hughes and Smakhtin [4] and Smakhtin and Masse [1] in the Introduction section, better highlighting the novelty of our work.

*R2: (1) Why select another gauged basin and use only its API, not its streamflow?*

For the German case study, we derived the streamflow and thus the FDC of each specific basin from the streamflow of another basin. We will better explain it in the text. We reported the evaluation metrics of the procedure to show the goodness of the method. We will better highlight the use of the streamflow as support variable.

*p.C2: (2) Why not use the API at the target basin?*

Because to retrieve the discharge at the target basin gauged during the target period, the discharge values only are necessary.

*R2: 2.a. The idea of using the reference gauge discharge is raised and in the first presentation of the methodology on page 10, the FDC at the reference basin is computed, but it is never used.*

AC: We will improve the presentation of the methodology and we will remove what is unnecessary to the development of the procedure.

*R2: 2.b. The results for the US basins are presented quite differently than those of the German basins, including using different performance criteria.*

AC: We decided to show results, in terms of flow duration curves, of the US catchments only to reduce the amount of possibly redundant data. In the revised version of the manuscript. we will use the same estimation metrics for both sites.

*R2: 2.c. How the performance criteria could be applied for individual predictions was not clear to me.*

AC: We will better explain it the text (Sect. 4.1) with the following text: "The **X** percentile is defined as the set containing all "**X**,…" numbers where the dots stand for the decimal points. For instance, the 1.09%, 1.36%, 1.63%, 1.91% belong with the 1rst percentile."

*R2: 2.d. The choice of basins for the study seems rather arbitrary. For example, there are hundreds of basins in the MOPEX, including many others that do not have much snow.*

AC: We decided to use these catchments also because the land use did not consistently change in the time window we used for analysis (see beginning of Sect. 3.2 (now Sect.3): "As resulted from the KS test, FDCs cannot be considered an invariant characteristic of a basin. The fact that FDCs are not invariant suggests that the weather is a driver of annual runoff variability. Indeed, the reason of the invariance must be the weather conditions as others (e.g. the catchment area, the land use) did not change.").

*R2: 2.e. The consideration of energy versus water limitation as a measure of similarity is interesting but it is not clear that it is relevant when API is being used.*

AC: In this paper the API is used to estimate the streamflow at a specific site. Annual streamflow variability is driven by the availability of water (i.e., provided by the precipitation) and energy (i.e., the evapotranspiration), [5]. Therefore, it is relevant to distinguish between water and energy limited catchments as their behavior is different.

*R2: 1.a. Are there snow effects in the Upper Neckar basin? How addressed?*

AC: Snow effects are considered by a simple snow accumulation and snowmelt model using a degree day approach. This allows to convert snow to a daily liquid water which is then used for the calculation of the API.

*R2: 1.b. Do you take the karstic effects on Upper Neckar flows into account?*

AC: For the karstic catchments a direct transfer seams not to be plausible. However the temporal stability of the API/Runoff can be considered as invariant. We'll check this assumption specifically.

*R2: 2.a.i. Figure 3: Perhaps plot and check correlation with temperature of ET/P instead of just ET?*

AC: The plot and the correlation of temperature and mean annual runoff (Q/P) already provide a similar information. It would be redundant.

R2: 2.a.ii. Figure 3: Need to consider uncertainty around correlation estimates: for the Peace R. it seems unlikely that rho = 0.027 is a significantly positive value; rather probably this basin is balanced between energy and water limitation by this criterion.

AC: We thank the reviewer for the suggestion, we will add the following consideration to the paper: "For instance, measurements at Peace River (LA) suggest that the catchment is balanced between energy and water limitation by the correlation criterion, Figure 3 upper panel."

R2: 2.b. Last sentence on p. 6: It is not possible…" It sounds plausible, but has this assertion been tested? The statement itself is very categorical; in fact there are degrees of water and energy limitation. How different do they need to be to make this true (if it is)? In particular, for the present application, the methodology may account for the water versus energy limitation; it may be that the timing of the weather is the most important thing to have in common.

AC: We tested the assertion. As a result it was not possible to estimate streamflow values of a water-limited catchment from the data of an energy-limited ones. See also reply to comment 2e.

R2: 3.a.i. Sect. 3.1. 2nd paragraph: It seems there should already be a well-established way of addressing autocorrelation effects on the K-S test.

AC: Weiss [6] proposed a methodology to account for modifying the K-S test for autocorrelated data. Later, Xu [7] suggested a method that can be applied to two sample test. However, our way to take into account for the autocorrelation is easier to implement and has a nice interpretation of equivalent sample size adjustment. More importantly, our method can be easily generalized to two samples test. We will introduce the following paragraph in the section: "[…] Since the streamflow data presents autocorrelation, the autocorrelation effects the KS test. Weiss (1978) proposed a methodology to account for modifying the K-S test for autocorrelated data. Later, Xu (2013) suggested a method that can be applied to two sample test. The information contained in the data is (usually) less than an i.i.d. sample with the same size. In other words, the number of equivalent independent observations is fewer than the sample size. In the following we explain how we took into account the equivalent sample size. It is easier to implement and more importantly, it can be easily generalized to two sample test. We can assume that the autocorrelation effect attenuates after three days [...]"

R2: 3.a.ii. a. Sect. 3.1. 2nd paragraph: The last two sentences of this paragraph seem to be referring to a test on a particular basin, but they are stated as if these relations are generally (i.e., mathematically) true. Which is it?

AC: This example is given for streamflow values at daily resolution recorded during a year, thus for a time series of 365 values. We will specify it with the following sentence in the paper: "For instance, let's take as an example the 1 year FDCs. If the samples were 3 times smaller and for instance their length would equal 122 …"

R2: 3.b. 3rd paragraph: This paragraph seems to include "Results", not "Methodology".

AC: In this paragraph we provide details about the methodology, but we also anticipate results regarding the KS test. This is done because those results explain the reasons why we applied the methodology presented in the following part of the paper to estimate the FDCs. For the sake of clarity, we decided to move the Sect. 3.1 in Sect. 2 and name it "2.4 Preliminary analysis".

*R2: 4.a. Section 3.2: First paragraph: It is not always true that the non-weather properties (land use) do not change. Did you check that your study basins satisfy this assumption?*

AC: We considered basins where the land use did not deeply change in time.

*R2: 4.b. Section 3.2: Last sentence: Did you test different values of alpha other than 0.85, or just select that value for the reasons given?*

AC: We tested also other values of alpha. However, we decided to proceed with alpha=0.85 because when α tends to 1, API represents the long memory of the basin as it includes the effect of precipitation occurred many days before. Moreover, this is in agreement with the value used by [8] for the Neckar Catchment. Therefore, we will add a line: "To capture this behaviour, in this study α is chosen equal to 0.85, this is in agreement with a previous study by Sugimoto (2014) who investigated the same case study area (i.e. Neckar catchment)."

*R2: 5.a Section 3.3. First paragraph, last sentence: Why do you assume "large scale precipitation"? What do you mean by that?*

AC: Small-scale variability of rainfall can be assumed to vary in a range lower than 10–20 km [9]. Therefore instead of APIs calculated from point precipitation areal precipitation is considered. The wording may be inappropriate and will be changed.

*R2: 5.b. Last complete paragraph on p. 10: It seems it would be better to interpolate between Pj and Pj+1 rather than taking the mean, but it may not make a lot of difference.*

AC: Thank you for the suggestion, as you anticipated, the difference is not significant.

*R2: 6.a. Section 4, p. 11, discussion of figures 5-8: a. Several statements regarding goodness of fit are made without being quantified. However the K-S technique has been presented and could be applied; indeed, it would be ideal to provide K-S test results to accompany the results in each panel of these plots.*

AC: Thank you for the suggestion, the KS distance D* will be added to each panel.

*R2: 7.a. Section 4, p. 13, figure 10 and discussion of it: a. Why present 30, 70, 90, and 99th percentiles? As one can see, 90th and 99th (though the lower right panel of figure 10 is labeled as the 95th percentile), are almost the same. The complementary percentiles, i.e., 70, 30, 10, and 1st percentiles (exceedance probabilities) would be more interesting, in my opinion.*

AC. We chose these percentiles as they are flow percentiles usually investigated in literature. For instance, the approach by Franchini and Suppo [10] regionalises these streamflow quantiles.

The title of the plots will be checked and made consistent with the caption.

*R2: 7.b. You say (lines 5-6 of p. 13): "it is not possible to estimate the flow quantiles using regression methods that do not take into account the weather characteristics." This may be an over-statement. You have demonstrated that if you want to transfer across time, weather fluctuations need to be considered. But for prediction at ungauged basins for a fixed period of time, that may not be true*

AC: The moving average is computed to show that the between-year variability of the discharge of a specific percentile can be high. Therefore, this suggests that percentiles cannot be considered an

invariant characteristic of the basin and thus they cannot be estimated using geographic and morphologic characteristic of the basin only.

This is true also for prediction at ungauged basins for a fixed period of time as we demonstrated applying the K-S test to streamflow values gauged during the same time window at two different sites (see Sect.3.1 lines 21-22).

*R2: 1. Section 3.1, 3rd paragraph: This paragraph seems to include "Results", not "Methodology.*

AC: In this paragraph we provide details about the methodology, but we also anticipated results regarding the KS test. This is done because the results justify the methodology we present in the paper to estimate the FDCs.

*R2: 2. Section 3.3, in steps 2&3 of the "procedure to predict" (p. 10), the FDCs of the reference catchment A is computed, but it does not seem to be used in the procedure.*

AC: We will remove it and rephrase the methodology section to make it clearer.

*R2: 3. Section 3.3, step 8 of the "procedure to predict" (p. 10): Suggest "qBrj is taken to be the value of discharge that occurred…" rather than simply "qBrj is the value of discharge that occurred…".*

AC: The Section will be rewritten to make it clearer.

*R2: 4. Section 3.3, last paragraph (p. 11): It is stated here that in the paper both discharge and precipitation will be used as the support variable. But everything before indicates that only precipitation will be used. And I don't see any results using discharge as the support variable.*

AC: For the German case study, the support variable is the discharge (please, see also lines 5-7 p.11). We will better explain it.

*R2: 5.a. Section 4, figures 5-8: From what period is this FDCref_site that is plotted? As it does not seem to be used in the procedure, why is it plotted?*

*5.b. Section 4, figures 5-8: I think however you should add the FDC of the target site during the reference period to these plots so the reader can see how much the FDC has changed from reference period to target period.*

AC: Since usually the FDC of a donor site is used to retrieve the FDC of a target site, the FDCref_site was plotted to show the difference between the FDC at the donor site and the FDC at the target site recorded during the same period of time. Each FDCref_site is recorded during the reference period reported either in the plot or in the caption.

*R2: 6.a.i. Section 4, figure 9 and discussion of it: Discussion of figure 9 on p. 12, lines 9&10: "Results shown that the distance between the former pairs is bigger than the distance between the latter, Figure 9.":i. I don't think you ever defined the K-S distance. That needs to be done.*

AC: The K-S distance will be defined as:

"Moreover, the test allows us to estimate the distance between couple of FDC:

$$D^* = \max_x \left( \left| F_1(x) - F_2(x) \right| \right), \tag{3}$$

where $F_1(x)$ is the proportion of $x_1$ values less than or equal to x and $F_2(x)$ is the proportion of $x_2$ values less than or equal to x. $F_1$ and $F_2$ are two FDCs."

*R2: 6.a.ii. Section 4, figure 9 and discussion of it: a. Discussion of figure 9 on p. 12, lines 9&10. I am willing to believe this assertion is true, but it is hard to see just from the plot. Can you provide some summary results such as the mean and median difference between 9 (top) and 9 (bottom) to give evidence of the assertion.*

AC: We will provide the following summary to evidence the findings: "Moreover, we wanted to know which is the distance between pairs of FDCs built at the same site in different periods of time. This distance is bigger than the distance between simulated and observed FDCs at the same period of time for the 73% of the cases, Figure 9."

*R2: 6.a.iii. Section 4, figure 9 and discussion of it: a. Discussion of figure 9 on p. 12, lines 9&10. This assertion should be restated without the shorthand of "former" and "latter". It is hard to understand the way it is currently phrased, and it is a very important point.*

AC: We will rephrase the sentence as explained in the comment above.

*R2: 7. Section 4, pp. 11-14: It is not clear why the Results section starts by giving a lot of results for the U.S. catchments and none for the German ones.*

AC: We decided to show results of U.S. in a comprehensive way (both FDCs and performance criteria are shown) to keep compact the manuscript avoiding redundant plots. For the German case study we shown the performance criteria which are much more representative than the FDCs. The performace criteria are shown in an extensive way as they are reported for both case studies.

*R2: 8.a. Section 4.1, pp. 14-17, Definition of performance criteria: Are all these computed for Q in mm units? Even though those are units used throughout, it would be worth re-emphasizing that here.*

AC: Yes, Q is in mm. We will better highlight it with this sentence "Discharge values are in mm."

*R2: 8.b. BIAS: This is not a simple bias as it is normalized by Qsim; it is more like a relative bias or "relative mean error"; however usually one divides by Qobs. Actually, ME (defined later) is more like a simple bias.*
*c. Why apply different criteria for the German catchments?*
*d. "Ratio":*
*i. Can you give it a more meaningful name?*
*ii. This formula looks odd. If the square root were only on the numerator, it would be the standard error divided by the mean error (and the quantity would be non-dimensional). But why apply the square root to the mean error in the denominator?*

AC: We thank you the reviewer for the suggestion, we will use the same metrics for both case study areas. We reviewed the estimation metrics, there was a typo in the BIAS formula, the correct form will be reported in the paper and below. Results were estimated with the following formula in agreement with Castellarin et al. [11]

$$\text{BIAS} = \frac{1}{N} - \sum_{i=1}^{N} \left( \frac{Q_{sim,i} - Q_{obs,i}}{Q_{obs,i}} \right).$$

*R2: 9. Section 4.1, figures 11-13: Many of the colors these figures are shifted so each box has more than one color, making them hard to interpret. This effect needs to be fixed.*

AC: We are sorry for this issue, we will fix it.

*R2: 10. Section 4.1: I don't see how the Performance criteria were applied to create the results shown in figures 11-13. As I understand, there is only one prediction of each quantile, for a fixed reference catchment and decade. Then how does one do the summations indicated in the performance criteria formulae? Following the definition of NSE on p. 15 it says: "N is the number of discharge values related to a specific percentile". How many of those are there? Is there ever more than one? If so, how? The situation with the correlation coefficient values presented in figure 14 seems to be the same: How does one compute correlation coefficients without multiple values? If there are multiple values, where are they coming from?*

AC: We will better explain it the text with the following sentence: "The **X** percentile is defined as the set containing all "**X**,…" numbers where the dots stand for the decimal points. For instance, the 1.09%, 1.36%, 1.63%, 1.91% belong with the 1rst percentile."

*R2: 11.a. Conclusions: p. 21, lines 2-5: "Here it is shown that two FDCs built for the same catchment, but with data corresponding to two different time windows, cannot be regarded as the same continuous distribution. The same results when two FDCs of two different catchments built for the same time window are analysed. Thus, it is not possible to infer a FDC using parameters retrieved from the distribution of another FDC without considering the weather." The first sentence supports the assertion in the third, but the second does not. If two different catchments experience possibly similar weather but produce a different streamflow, the cause is not the weather.*

AC: We thank the reviewer for the suggestion. The sentences will be: "Here it is shown that two FDCs built for the same catchment, but with data corresponding to two different time windows, cannot be regarded as the same continuous distribution. It is not possible to infer a FDC using parameters retrieved from the distribution of another FDC without considering the weather.

*R2: 11.b. Conclusions: p. 21, lines 13-14: "Since precipitation data series are characterized by a high number of zeros, here we used the Antecedent Precipitation Index (API)." This statement misses the more important fact that the API combines in a streamflow-like way the history of the precipitation. (A similar statement is made at the beginning of section 3.2 near the bottom of p. 8.)*

AC: We thank the reviewer for the suggestion. We will rephrase as:

"Since precipitation data series are characterized by a high number of zeros, here we used the Antecedent Precipitation Index (API). The API is used as it represents in a streamflow-like way the precipitation of the basin. It represents the memory of a basin as it provides the amount of precipitation released by the soil throughout the time."

*R2: 11.c. p. 22, lines 26-27: Qualitative statement about similarity in shape from beginning of Section 4 is repeated. This assertion needs to be quantified somehow.*

AC: A quantitative assessment of the goodness of the methodology is performed through the performance criteria. To describe why there can be a difference between the two FDCs, we will rephrase the sentence as:

"The distance between the simulated and observed FDCs can be due to the different temperature values characterizing the two catchments."

**Author's changes in manuscript**

Some hints regarding authors' changes in the manuscript have been already given in the comments section. In the following, a summary of authors' changes in manuscript based on comments of all referees is given:

1. In the Introduction, we will clarify the novelty and the contribution of the paper.
2. The Methodology section will be reorganized and improved to provide a clearer description of the method. For the sake of clarity, a figure will be added.
3. We will provide a clear statement of the hypothesis.
4. The sections will be organized on the base of the Introduction-Methods-Results-and-Discussion (IMRAD) format.
5. For both case studies we will use the same performance criteria and results will be discussed in-depth.

**References**

[1]    V.Y. Smakhtin, B. Masse, Continuous daily hydrograph simulation using duration curves of a precipitation index, Hydrol. Process. 14 (2000) 1083–1100. doi:10.1002/(SICI)1099-1085(20000430)14:6<1083::AID-HYP998>3.0.CO;2-2.

[2]    V.Y. Smakhtin, D.A. Hughes, E. Creuse-Naudin, Regionalization of daily flow characteristics in part of the Eastern Cape, South Africa, Hydrol. Sci. J. 42 (1997).

[3]    V.Y. Smakhtin, Generation of natural daily flow time-series in regulated rivers using a non-linear spatial interpolation technique, Regul. Rivers Res. Manag. 15 (1999) 311–323.

[4]    D.A. Hughes, V. Smakhtin, Daily flow time series patching or extension: a spatial interpolation approach based on flow duration curves, Hydrol. Sci. J. 41 (1996) 851–871. doi:10.1080/02626669609491555.

[5]    G. Blöschl, M. Sivapalan, W. Thorsten, A. Viglione, H. Savenije, Runoff prediction in ungauged basins: Synthesis across Processes, Places and Scales, Cambridge University Press, 2013.

[6]    M.S. Weiss, Modification of the Kolmogorov-Smirnov statistic for use with correlated data, J. Am. Stat. Assoc. 73 (1978) 872–875. doi:10.1080/01621459.1978.10480116.

[7]    X. Xu, Methods in Hypothesis Testing, Markov Chain Monte Carlo and Neuroimaging Data Analysis, 2014. https://dash.harvard.edu/handle/1/11108711.

[8]    T. Sugimoto, Copula based Stochastic Analysis of Discharge Time Series, 2014.

[9]    D.K. Nykanen, E. Foufoula-Georgiou, W.M. Lapenta, Impact of Small-Scale Rainfall Variability on Larger-Scale Spatial Organization of Land–Atmosphere Fluxes, J. Hydrometeorol. 2 (2001) 105–121.

[10]   M. Franchini, M.Suppo, Regional analysis of flow duration curves for a limestone region, Water Resour. Manag. 10 (1996) 199–218.

[11]   A. Castellarin, D. Burn, A. Brath, Assessing the effectiveness of hydrological similarity measures for flood frequency analysis, J. Hydrol. 241 (2001) 270–285. doi:10.1016/S0022-

1694(00)00383-8.

---

## Author Response (AR1)

We would like to thank William Farmer for the time dedicated to our paper and for his valuable comments that contribute to improve the paper.

In the following, reviewer's comments are in Italic (R2), Authors' comments are in normal text (AC). Some of the changes made in the manuscript by the Authors based on Referee's comments are reported below. Moreover, in the following, a summary of authors' changes in manuscript based on comments of the editor and all referees is given:

1. In the Introduction, we added a discussion on the works by Hughes and Smakhtin (1996) and Smakhtin and Masse (2000) clarifying the differences existing between our and their work.
2. The Methodology section was reorganized and improved to provide a clear description of the method. Moreover, for the sake of clarity, a figure was added.
3. We provided a clear statement of the hypothesis.
4. The sections are now organized on the base of the Introduction-Methods-Results-and-Discussion (IMRAD) format.
5. For both case studies we have used the same performance criteria and results are discussed in-depth.

*R1: This work could benefit from a clear statement of hypotheses. In my opinion, the main hypothesis is that the daily flow duration curves at an ungauged location can be simulated with knowledge of the precipitation record at both the ungauged site and some index site. This hypothesis relies on a further assumption that the cumulative distributions of streamflow and precipitation correlate in some way. In the revisions I am proposing, I think the authors should clearly set out to quantifiably address these hypotheses. A figure may improve the understanding of the methodology.*

**AC: We thank the reviewer for the suggestion, we stated the hypothesis of the work (see comments below) and added a figure to give a clearer explanation of the procedure.**

*R1: As a previous commenter noted, the approach is difficult to understand and may be greatly simplified. The authors create cumulative distribution functions (CDFs) for streamflow and API at both an index site and a target site over some reference period. They then create a CDF of streamflow and API at the index site for some target period and a CDF for API at the target site for this same target period. The method then only uses the CDFs of (1) API at the index site in the target period, (2) API at the index site in the reference period, and (3) streamflow at the target site in the reference period.*

**AC: We thank the reviewer for the comment, we changed Sect. 3 to make it clearer and shorter.**

*R1: In addition to improving readability, revising the methods section might also address the concerns raised by the previous commenter. Namely, that it seems the approach could be greatly simplified through interpolation along the relevant relationships without the need for intermediate*

*exceedance probabilities. This would be accomplished by the following: (1) Create the CDF of API at the index site in the target period, (2) Plot the API of the index site in the reference period against the streamflow at the target site during the reference period, (3) interpolate each API from the target period, in order, along the curve created in (2) to produce the CDF of streamflow at the target site in the target period. While this approach is achievable algorithmically, and identical to the one proposed, it raises several concerns about the implicit assumptions. "Are API and streamflow ranked independently? (See step 3 of page 10.) If so, the implicit assumption is that the exceedance probability of the API on a given day is equivalent to the exceedance probability of the streamflow on that same day at the same site. This is a pretty sizeable assumption. As a start, it would be good to see if the temporal sequence of API exceedance probabilities is highly correlated with the temporal sequence of streamflow exceedance probabilities at a single site over the same period."*

AC: We thank the reviewer for the comment, the methodology section was largely modified and a clear statement of the hypothesis is provided. Specifically, it is also shown that the temporal sequence of API exceedance probabilities is highly correlated with the temporal sequence of streamflow exceedance probabilities at a single site over the same period. For instance, in the manuscript the following table showing this correlation is reported for three catchments for four different periods of time.

| | **Correlation** | | |
|---|---|---|---|
| **Period** | Blanco | Tangipahoa | Choctawthachee |
| **1948-1968** | 0.978 | 0.996 | 1 |
| **1968-1988** | 0.995 | 0.997 | 1 |
| **1948-1963** | 0.998 | 0.993 | 0.998 |
| **1948-1958** | 0.970 | 0.995 | 0.998 |

*R1: The second assumption arises when we move from the CDF of API at the index site in the reference period to the CDF of streamflow at the target site in the reference period. This movement introduces a second implicit assumption: namely, that the exceedance probability of the API on a given day at the index site is equivalent to the exceedance probability of streamflow on that same day at the target site. Put another way, if you accept the assumption in the previous paragraph, this step assumes that the temporal sequencing of API is identical at both sites. Again, this needs to be demonstrated: Is the temporal sequence of API exceedance probabilities at the index site highly correlated with the temporal sequence of streamflow exceedance probabilities at the target site in the reference period? It may be argued that the temporal sequencing is irrelevant. This is not the case. By assuming the same exceedance probabilities in step 7 of page 10, we are assuming a perfect correlation and, therefore, assuming a temporal correspondence.*

AC: We thank the reviewer for the comment, in the revised manuscript we shown that the temporal sequence of API exceedance probabilities at the index site is highly correlated with the temporal sequence of streamflow exceedance probabilities at the target site in the reference period. As an example, the following table was added to the paper to report the correlation between Blanco (index site) and three target sites (specified in the table) for four different reference periods.

| Sites | 1948-1968 | 1968-1988 | 1948-1963 | 1948-1958 |
|---|---|---|---|---|
| **Tangipahoa** | 0.996 | 0.997 | 0.994 | 0.997 |
| **Choctwhatchee** | 1 | 1 | 0.999 | 0.999 |
| **Bogue** | 0.990 | 0.992 | 0.995 | 1 |

*R1: The third assumption, which was alluded to earlier, is that the CDF of API is identical across sites for both the index site and the target site in the same period. This is what allows the authors to use the CDF of the API of the index site in the target period for step 6 on page 10. It may be that this is what the authors meant by "the assumption of large scale precipitation" (line 16, page 9); if so, please clarify. Regardless, this assumption needs to be validated through correlation or a KS test.*

AC: We thank the reviewer for the comment, in the revised manuscript we now show that the CDF of API is identical across sites for both the index site and the target site in the same period through correlation. In the table below an example of correlation between Blanco (USA) and three other sites is reported for four different time periods.

| Sites | 1948-1968 | 1968-1988 | 1948-1963 | 1948-1958 |
|---|---|---|---|---|
| **Tangipahoa** | 0.98 | 0.99 | 0.97 | 0.99 |
| **Choctwhatchee** | 0.98 | 0.98 | 0.98 | 0.98 |
| **Bogue** | 0.99 | 0.99 | 0.98 | 0.99 |

Moreover, the distributions of API have the same type of distribution – Weibull is accepted for all of them. On the other hand, the distribution parameters may differ from site to site and from time period to time period. For instance, the Weibull is the best fitting distribution of the API at Blanco for the periods 1948-1968, 1968-1988, 1948-1963 and 1948-1958. The same applies to the API of the three sites above for the same periods. We now specify it in the paper as well.

*R1: Without some quantifiable validation of these assumptions, the proposed method is tenuous at best and left vulnerable to criticism. With that in mind, and the comments of the previous commenter, I'd like to propose that exploring these assumptions might result in modifications of the methodology that might move away from the case of simple interpolation. Is the relationship between API and streamflow constant across periods or sites? Should API and streamflow be ranked independently or with some sort of dependence? Should the API of the index site in the target period be used to map to a different site in a different period (i.e., the target site in the reference period)? Exploring these questions, and validating the underlying assumptions, will produce a more robust approach."*

AC: Thank you for this comment. We assume that the relationship between API and FDC is the same for the same site regardless the time period. For the reviewer we enclose the following table reporting the correlation at Blanco between API and q for different time periods. This correlation is stronger than the one obtained considering API and discharge from two different time periods.

| Site | 1948-1968 | 1968-1988 | 1948-1963 | 1948-1958 |
|---|---|---|---|---|
| **Blanco (USA)** | 0.977 | 0.995 | 0.997 | 0.970 |

*R1: "In addition to their main hypotheses proposing this methodology, the authors assert that the FDC is a product of the basin and the weather. This is surely intuitive, but the evidence provided could be greatly strengthened. The authors use KS tests, but is unclear how they were applied. It would be informative to clearly communicate if the CDF of streamflow from one period and the CDF of streamflow from another period could be considered significantly different. The authors have done this, but the presentation is not clear. The extension would be to ask if the API can be correlated with any differences across time. (As an aside: Was there any discussion of selecting stationary sites? How would nonstationary behavior play a role here?)"*

AC: The FDC seems to be significantly different from one time period to the other. The same applies to the API. In our opinion, this is not caused by non stationarity of the time series but more to some long memory effects. We show that this long scale variability is very visible in the discharge.

*R1: This, in my opinion, raises another concern: The authors seem to be attempting to simultaneously address two very different problems. The first problem considers a target site that has a streamflow record overlapping with an index site, but the desired period has no overlap (the ungauged area is the same site, different period). In this case, the use of APIs within site, without an index site, would be most ideal. The second problem considers a site without any streamflow information; this situation necessitates the use of an index. Of course, when there are gaps in the API record as well, this transforms into four unique problems. Regardless, if we believe the underlying assumption that the CDF of streamflow is a product of basin and weather, then the solutions to these problems must be quite different. The first asks if knowledge of new weather can produce the CDF of streamflow, while the second alters both variables and asks if the CDF relationship can be transferred across weather and basin. Line 8 of page 3 implies that both problems are considered, but the remainder of the paper seems only to address the partially gauged site. I would advise addition of the second problem or, at least, a discussion of implication for the second problem (completely ungauged).*

AC: We thank the reviewer for the comment, the methodology explained in the paper needs to be applied to partially gauged basins, we eliminated the reference to totally ungauged sites in the paper.

*R1: In 1996, Hughes and Smakhtin (<https://doi.org/10.1080/02626669609491555>), among others, provided a technique for hydrograph simulation using flow duration curves. While their focus was on hydrographs, the extensions to ungauged FDCs can be made quite clearly (i.e., they could be derived from simulated hydrographs). Smakhtin and Masse (2000: <https://doi.org/10.1002/(SICI)1099-1085(20000430)14:6%3C1083::AID-HYP998%3E3.0.CO;2-2>) then extended this method to use a precipitation index. While I believe that the methods presented here are different, the novelty of this new method must be strongly articulated.*

AC: In the new version of the manuscript we recalled the two papers in the Introduction section. We added a deep discussion on the assumptions underlying their work and highlighted which are the differences of our work.

*R1: "I strongly encourage the authors to revisit the style of the manuscript. At times, it feels a bit disjointed and it may be improved by enforcing a strict Introduction-Methods-Results-and-Discussion (IMRAD) format. For example, section 3.1 is ostensibly a methods sections but presents a series of results that I think are pivotal to the paper (line 19, page 8). Similarly, the paragraph on page 13 and section 4.1 present new methods of analysis that have not been presented earlier in the paper. While*

*IMRAD is not a requirement, I do suggest thinking carefully about the best approach to presenting the narrative."*

AC: We thank the reviewer for the suggestion, we performed a restyle of the paper. The results reported in Sect.3.1 are pivotal to the development of the paper, they are now moved under a paragraph named "Preliminary analysis". The presentation of the performance criteria was moved into the Methodology

*R1: "In my opinion, this work needs more presentation and discussion of quantified results.*

AC: In the updated version of the paper, we have further developed the discussion.

*R1: "The results sections heavily rely on visualization. Even the presentation of metrics in section 4.1 is visual. While this is useful, we still need to see some discussion on the performance metrics. For example, the scale on NSE in Figure 11 makes all positive values appear as a single color. This presentation means we can't honestly see how the methods perform."*

AC: We chose to present results in a visual style to better show them. Because of the large number of sites and the large number of time windows we have investigated, to a reader it would take too much time to go through a tabular presentation, while plots have an immediate impact. However, we agree with the reviewer regarding the scale of the plots that sometimes make difficult to understand the goodness of the results. We have now solved this issue.

*R1: Page 1, line 17: When talking about general duration curves, more commonly known as cumulative distribution functions, it is better to say "exceedance frequency" rather than "exceedance time".*

AC: We replaced the word as suggested.

*R1: Page 2, line 1: Please provide the citation for the Weibull plotting position.*

AC: The citation was added (i.e., Weibull, W., 1939: A statistical theory of the strength of materials. Ing. Vetensk. Akad. Handl., 151, 1–45).

*R1: Page 3, line 4: Please provide more discussion and literature of this important point.*

AC: The sentence was referring to the results of the paper anticipating them for the readers. We rephrased the paragraph in the Introduction and we have introduced this concept in the Conclusions section together with an explanation.

*R1: Page 3, line 7: It is not clear what the "distribution of the FDC" is. The FDC is a distribution, so it is confusing to talking about the distribution of a distribution.*

AC: We rephrased the sentence as "It is not possible to develop relations between parameters of the basin and characteristics of the FDC to yield synthesized FDCs in locations where flow data are not available, as done for instance by Quimpo et al. [5]."

*R1: Page 4, line 6: Florida, Louisiana and Texas are certainly not the East coast. I would suggest the Gulf Coast.*

AC: We rephrased the sentence

*R1: Page 5, line 7: misspelling of database*

AC: We carefully checked the English spelling throughout the paper.

*R1: Page 5, line 2 (?): This is an example of inconsistent citation style. Bloeschl should be in parenthesis.*

AC: We carefully checked the citation style throughout the paper.

*R1: Page 8, line 5: Please provide citation to KS test.*

AC: We provided the citation (i.e. Massey, F. J. The Kolmogorov-Smirnov Test for Goodness of Fit. Journal of the American Statistical Association. Vol. 46, No. 253, 1951, pp. 68–78.).

*R1: Page 9, line 10: What lead to this choice for alpha? (Also, note that the same symbol is used earlier in this section for significance: page 8, line 10.) Please provide citation or summary of initial exploration.*

AC: Now $\alpha$ refers to the API only. We specified in the paper that when $\alpha$ tends to zero, API keeps tracks of the precipitation occurred in the few previous days and it represents the short memory of the basin. When $\alpha$ tends to 1, API represents the long memory of the basin as it includes the effect of precipitation occurred many days before. To capture this behavior, in this study $\alpha$ is chosen equal to 0.85. Moreover, this is in agreement with a previous study [6] which investigated the same case study area (i.e. Neckar catchment).

*R1: Page 9, line 18: I strongly suggest referring to the "reference site" as an "index" or "donor". The reference connotation implies lack of human influence that might be confusing. The same could be said of the reference period*

AC: We are now using "donor" site instead of "reference" one.

*R1: Page 10, line 17: The series of Nr and Nt are both being indexed with i, which leads to confusion.*

AC: Please notice that we consistently revised the Methodology section, however we changed the Appendix where both $Nt$ and $Nr$ exist.

*R1: Page 10, line 19: So, API_Ati is equal to API_Arj?*

AC: Yes, it is. We added the following Figure to better clarify the methodology and also we re-organized and strongly improved the methodology section.

[Figure]

R1: Page 11, line 4: What is a supporting variable? This is not described as such earlier, so it surprises the reader.

AC: The supporting variable is the one used to retrieve the streamflow values, in this example is the API. For the sake of clarity, we have now called it "proxy" variable.

R1: Page 11, line 17: "good agreement" is very subjective. Please provide thorough, quantifiable analysis. For example, a lot of the curves in figure 6 look rather poor for highs and lows (top row, second box from the left).

AC: The sentence is an introduction to the extensive explanation presented in the following lines of the manuscript. The paragraph reporting the results in terms of performance measures is more extended and further shows how the method has a higher performance per intermediate flows, while it is poorer for high and low flows. Nevertheless, the error in terms of Mean Absolute Error is small also for high and low flows, showing an overall good performance of the approach also for extreme flows.

R1: Figure 5: Why was the box for ref:68-88 and tar: 88-98 not included? The caption needs to do a better job of describing the different panels.

AC: The missing panel would not add more information to the paper, therefore because of the lack of space the panel was not included. We better described the panels in the caption.

R1: Page 12, line 9: Spelling of FDCs

AC: We carefully checked the spelling throughout the paper.

R1: Page 13: The methods for this paragraph were very unclear to me. Could a figure or a revision help?

AC: In this paragraph we compute the moving average to show that the between-year variability of the discharge of a specific percentile can be high. Therefore, this suggests that percentiles cannot be considered an invariant characteristic of the basin and thus they cannot be estimated using geographic and morphologic characteristic of the basin only. We decided to show the moving averages of these specific percentiles as they are the most used ones.

*R1: Page 14: Please provide the citation for NSE. Even better, a metric like KGE might be more appropriate*

AC: The citation was added. We used the NSE, despite the criticism it has received (e.g.,[7]) because of the familiarity most hydrologists and meteorologists have with it [8], facilitating the interpretation of the obtained values.

*R1: Figure 10: What is the horizontal axis of this figure?*

AC: Figure 10 was replaced with a revised figure

*R2: This paper presents a method for estimating FDCs during an ungauged period at a "target" location that is gauged during another period. [...].As such it is similar to a record extension application of the approach of Smakhtin and Masse (2000, Hydrological Processes, Vol. 14, pp. 1083-1100), except they estimated daily flow in ungauged basins, and to the work of Hughes and Smakhtin [...] Ideally, however, the authors would investigate the distinction and show how in application one might make different choices regarding parameters or selection of the reference basin when estimating the FDC as opposed to daily streamflow.*

AC: In the new version of the manuscript, we recalled the two papers in the Introduction section. We added a deep discussion on the assumptions underlying their work and highlighted which are the differences of our work.

*R2: (1) Why select another gauged basin and use only its API, not its streamflow?*

AC: For the German case study, we derived the streamflow and thus the FDC of each specific basin from the streamflow of another basin, this was now better specified in the text. We reported the evaluation metrics of the procedure to show the goodness of both methods.

*p.C2: (2) Why not use the API at the target basin?*

Because to retrieve the discharge at the target basin gauged during the target period, the API is not necessary at the target site.

*R2: 2.a. The idea of using the reference gauge discharge is raised and in the first presentation of the methodology on page 10, the FDC at the reference basin is computed, but it is never used.*

AC: We improved the presentation of the methodology and removed what is unnecessary to the development of the procedure.

*R2: 2.b. The results for the US basins are presented quite differently than those of the German basins, including using different performance criteria.*

AC: We decided to show results, in terms of flow duration curves, of the US catchments only to reduce the amount of possibly redundant data. In the revised version of the manuscript, we used the same estimation metrics for both sites.

*R2: 2.c. How the performance criteria could be applied for individual predictions was not clear to me.*

AC: We explained it the text (Sect. 3.3) with the following text: "The **X** percentile is defined as the set containing all "**X**,…" numbers where the dots stand for the decimal points. For instance, the 1.09%, 1.36%, 1.63%, 1.91% belong with the 1rst percentile."

*R2: 2.d. The choice of basins for the study seems rather arbitrary. For example, there are hundreds of basins in the MOPEX, including many others that do not have much snow.*

AC: We decided to use these catchments also because the land use did not consistently change in the time window we used for analysis.

*R2: 2.e. The consideration of energy versus water limitation as a measure of similarity is interesting but it is not clear that it is relevant when API is being used.*

AC: Annual streamflow variability is driven by the availability of water (i.e., provided by the precipitation) and energy (i.e., the evapotranspiration), [5]. The API combines the precipitation to provide the amount of the water released by the soil. Therefore, it is relevant to distinguish between water and energy limited catchments, as their behavior is different.

*R2: 1.a. Are there snow effects in the Upper Neckar basin? How addressed?*

AC: Snow effects are considered by a simple snow accumulation and snowmelt model using a degree day approach. This allows to convert snow to a daily liquid water which is then used for the calculation of the API. We specified it in the text.

*R2: 1.b. Do you take the karstic effects on Upper Neckar flows into account?*

AC: For the karstic catchments, a direct transfer seams not to be plausible. However, the temporal stability of the API/Runoff can be considered as invariant.

*R2: 2.a.i. Figure 3: Perhaps plot and check correlation with temperature of ET/P instead of just ET?*

AC: The plot and the correlation of temperature and mean annual runoff (Q/P) already provide a similar information. It would be redundant.

*R2: 2.a.ii. Figure 3: Need to consider uncertainty around correlation estimates: for the Peace R. it seems unlikely that rho = 0.027 is a significantly positive value; rather probably this basin is balanced between energy and water limitation by this criterion.*

AC: We thank the reviewer for the suggestion, we added the following consideration to the paper: "For instance, measurements at Peace River (LA) suggest that the catchment is balanced between energy and water limitation by the correlation criterion, Figure 3 upper panel."

*R2: 2.b. Last sentence on p. 6: It is not possible…" It sounds plausible, but has this assertion been tested? The statement itself is very categorical; in fact there are degrees of water and energy limitation. How different do they need to be to make this true (if it is)? In particular, for the present application, the methodology may account for the water versus energy limitation; it may be that the timing of the weather is the most important thing to have in common.*

AC: We tested the assertion. As a result it was not possible to estimate streamflow values of a water-limited catchment from the data of an energy-limited ones. See also reply to comment 2e (above).

*R2: 3.a.i. Sect. 3.1. 2nd paragraph: It seems there should already be a well-established way of addressing autocorrelation effects on the K-S test.*

AC: Weiss [6] proposed a methodology to account for modifying the K-S test for autocorrelated data. Later, Xu [7] suggested a method that can be applied to two sample test. However, our way to take into account for the autocorrelation is easier to implement and has a nice interpretation of equivalent sample size adjustment. More importantly, our method can be easily generalized to two sample test. We introduced the following paragraph in the section: "… Since the streamflow data presents autocorrelation, the autocorrelation effects the KS test. Weiss (1978) proposed a methodology to account for modifying the K-S test for autocorrelated data. Later, Xu (2013) suggested a method that can be applied to two sample test. The information contained in the data is (usually) less than an i.i.d. sample with the same size. In other words, the number of equivalent independent observations is fewer than the sample size. In the following, we explain how we took into account the equivalent sample size. It is easier to implement and more importantly, it can be easily generalized to two sample test. We can assume that the autocorrelation effect attenuates after three days..."

*R2: 3.a.ii. a. Sect. 3.1. 2nd paragraph: The last two sentences of this paragraph seem to be referring to a test on a particular basin, but they are stated as if these relations are generally (i.e., mathematically) true. Which is it?*

AC: This example is given for streamflow values at daily resolution recorded during a year, thus for a time series of 365 values. We specified it with the following sentence in the paper: "For instance, let's take as an example the 1 year FDCs. If the samples were three times smaller and for instance their length would equal 122 …"

*R2: 3.b. 3rd paragraph: This paragraph seems to include "Results", not "Methodology".*

AC: In this paragraph, we provide details about the methodology, but we also anticipate results regarding the KS test. This is done because those results explain the reasons why we applied the methodology presented in the following part of the paper to estimate the FDCs. For the sake of clarity, we decided to move the Sect. 3.1 in Sect. 2 and name it "2.4 Preliminary analysis".

*R2: 4.a. Section 3.2: First paragraph: It is not always true that the non-weather properties (land use) do not change. Did you check that your study basins satisfy this assumption?*

AC: We considered basins where the land use did not deeply change in time.

*R2: 4.b. Section 3.2: Last sentence: Did you test different values of alpha other than 0.85, or just select that value for the reasons given?*

AC: We tested also other values of alpha. Then, we decided to proceed with alpha=0.85 because when α tends to 1, API represents the long memory of the basin as it includes the effect of precipitation occurred many days before. Moreover, this is in agreement with the value used by [8] for the Neckar Catchment. Therefore, we added a line: "To capture this behavior, in this study α is chosen equal to 0.85, this is in agreement with a previous study by Sugimoto (2014) who investigated the same case study area (i.e. Neckar catchment)."

*R2: 5.a Section 3.3. First paragraph, last sentence: Why do you assume "large scale precipitation"? What do you mean by that?*

AC: Small-scale variability of rainfall can be assumed to vary in a range lower than 10–20 km [9]. Therefore instead of APIs calculated from point precipitation areal precipitation is considered. The wording may be inappropriate and was changed.

*R2: 5.b. Last complete paragraph on p. 10: It seems it would be better to interpolate between Pj and Pj+1 rather than taking the mean, but it may not make a lot of difference.*

AC: Thank you for the suggestion, as you anticipated, the difference is not significant.

*R2: 6.a. Section 4, p. 11, discussion of figures 5-8: a. Several statements regarding goodness of fit are made without being quantified. However the K-S technique has been presented and could be applied; indeed, it would be ideal to provide K-S test results to accompany the results in each panel of these plots.*

AC: Thank you for the suggestion, the KS distance D* was added to each panel.

*R2: 7.a. Section 4, p. 13, figure 10 and discussion of it: a. Why present 30, 70, 90, and 99th percentiles? As one can see, 90th and 99th (though the lower right panel of figure 10 is labeled as the 95th percentile), are almost the same. The complementary percentiles, i.e., 70, 30, 10, and 1st percentiles (exceedance probabilities) would be more interesting, in my opinion.*

AC. We chose these percentiles as they are flow percentiles usually investigated in literature. For instance, the approach by Franchini and Suppo [10] regionalises these streamflow quantiles.

The title of the plots is now consistent with the caption.

*R2: 7.b. You say (lines 5-6 of p. 13): "it is not possible to estimate the flow quantiles using regression methods that do not take into account the weather characteristics." This may be an over-statement. You have demonstrated that if you want to transfer across time, weather fluctuations need to be considered. But for prediction at ungauged basins for a fixed period of time, that may not be true*

AC: The moving average is computed to show that the between-year variability of the discharge of a specific percentile can be high. Therefore, this suggests that percentiles cannot be considered an invariant characteristic of the basin and thus they cannot be estimated using geographic and morphologic characteristic of the basin only.

This is true also for prediction at ungauged basins for a fixed period of time as we demonstrated applying the KS test to streamflow values gauged during the same time window at two different sites (see Sect.2.4).

*R2: 1. Section 3.1, 3rd paragraph: This paragraph seems to include "Results", not "Methodology.*

AC: In this paragraph we provide details about the methodology, but we also anticipated results regarding the KS test. This is done because the results justify the methodology we present in the paper to estimate the FDCs. This paragraph is now in a new one named "Preliminary analysis".

*R2: 2. Section 3.3, in steps 2&3 of the "procedure to predict" (p. 10), the FDCs of the reference catchment A is computed, but it does not seem to be used in the procedure.*

AC: The methodology section is now revised.

*R2: 3. Section 3.3, step 8 of the "procedure to predict" (p. 10): Suggest "qBrj is taken to be the value of discharge that occurred…" rather than simply "qBrj is the value of discharge that occurred…".*

AC: The methodology section is now revised.

*R2: 4. Section 3.3, last paragraph (p. 11): It is stated here that in the paper both discharge and precipitation will be used as the support variable. But everything before indicates that only precipitation will be used. And I don't see any results using discharge as the support variable.*

AC: For the German case study, the support variable is the discharge. We better highlighted it, moreover the support variable is now recalled as "proxy" variable.

*R2: 5.a. Section 4, figures 5-8: From what period is this FDCref_site that is plotted? As it does not seem to be used in the procedure, why is it plotted?*

*R2: 5.b. Section 4, figures 5-8: I think however you should add the FDC of the target site during the reference period to these plots so the reader can see how much the FDC has changed from reference period to target period.*

AC: Since usually the FDC of a donor site is used to retrieve the FDC of a target site, the FDCref_site was plotted (now called FDC_donor site) to show the difference between the FDC at the donor site and the FDC at the target site recorded during the same period of time.

*R2: 6.a.i. Section 4, figure 9 and discussion of it: Discussion of figure 9 on p. 12, lines 9&10: "Results shown that the distance between the former pairs is bigger than the distance between the latter, Figure 9.":i. I don't think you ever defined the K-S distance. That needs to be done.*

AC: The K-S distance is now defined as:

"Moreover, the test allows us to estimate the distance between couple of FDC:

$$D^* = \max_x \left( \left| F_1(x) - F_2(x) \right| \right),$$

where $F_1(x)$ is the proportion of $x_1$ values less than or equal to x and $F_2(x)$ is the proportion of $x_2$ values less than or equal to x. $F_1$ and $F_2$ are two FDCs."

*R2: 6.a.ii. Section 4, figure 9 and discussion of it: a. Discussion of figure 9 on p. 12, lines 9&10. I am willing to believe this assertion is true, but it is hard to see just from the plot. Can you provide some summary results such as the mean and median difference between 9 (top) and 9 (bottom) to give evidence of the assertion.*

AC: We now provide the following summary to evidence the findings: "On the contrary, the test rejected the null hypothesis that FDCs built at the same location in different periods had the same distribution. In the 73% of the cases, the distance between pairs of interpolated and observed FDCs of the same period is smaller than the distance between FDCs built at the same site from data recorded during different periods, Figure 13."

*R2: 6.a.iii. Section 4, figure 9 and discussion of it: a. Discussion of figure 9 on p. 12, lines 9&10. This assertion should be restated without the shorthand of "former" and "latter". It is hard to understand the way it is currently phrased, and it is a very important point.*
AC: We rephrased the sentence "In the 73% of the cases, the distance between pairs of interpolated and observed FDCs of the same period is smaller than the distance between FDCs built at the same site from data recorded during different periods, Figure 13. These results suggest that the methodology proposed here has a good performance and it is actually an interesting alternative to other methodologies, which assume that FDC of different periods of time have the same distribution."

*R2: 7. Section 4, pp. 11-14: It is not clear why the Results section starts by giving a lot of results for the U.S. catchments and none for the German ones.*

AC: We decided to show results of U.S. in a comprehensive way (both FDCs and performance criteria are shown) to keep compact the manuscript avoiding redundant plots. For the German case study, we shown the performance criteria which are much more representative than the FDCs. The performance criteria are shown in an extensive way as they are reported for both case studies.

*R2: 8.a. Section 4.1, pp. 14-17, Definition of performance criteria: Are all these computed for Q in mm units? Even though those are units used throughout, it would be worth re-emphasizing that here.*

AC: Yes, Q is in mm. We specified it.

*R2: 8.b. BIAS: This is not a simple bias as it is normalized by Qsim; it is more like a relative bias or "relative mean error"; however usually one divides by Qobs. Actually, ME (defined later) is more like a simple bias.*
*c. Why apply different criteria for the German catchments?*
*d. "Ratio":*
*i. Can you give it a more meaningful name?*
*ii. This formula looks odd. If the square root were only on the numerator, it would be the standard error divided by the mean error (and the quantity would be non-dimensional). But why apply the square root to the mean error in the denominator?*

AC: We thank the reviewer for the suggestions, we used the same metrics for both case study areas. We reviewed the estimation metrics, there was a typo in the BIAS formula, the correct form is now reported in the paper and below. Results were estimated with the following formula in agreement with Castellarin et al. [11]

$$\text{BIAS} = \frac{1}{N} - \sum_{i=1}^{N} \left( \frac{Q_{sim,i} - Q_{obs,i}}{Q_{obs,i}} \right).$$

*R2: 9. Section 4.1, figures 11-13: Many of the colors these figures are shifted so each box has more than one color, making them hard to interpret. This effect needs to be fixed.*

AC: We are sorry for this issue, we fixed it.

*R2: 10. Section 4.1: I don't see how the Performance criteria were applied to create the results shown in figures 11-13. As I understand, there is only one prediction of each quantile, for a fixed reference catchment and decade. Then how does one do the summations indicated in the performance criteria formulae? Following the definition of NSE on p. 15 it says: "N is the number of discharge values related to a specific percentile". How many of those are there? Is there ever more than one? If so, how? The situation with the correlation coefficient values presented in figure 14 seems to be the same: How does one compute correlation coefficients without multiple values? If there are multiple values, where are they coming from?*

AC: We better explained it the text with the following sentence: "The **X** percentile is defined as the set containing all "**X**,…" numbers where the dots stand for the decimal points. For instance, the 1.09%, 1.36%, 1.63%, 1.91% belong with the 1rst percentile."

*R2: 11.a. Conclusions: p. 21, lines 2-5: "Here it is shown that two FDCs built for the same catchment, but with data corresponding to two different time windows, cannot be regarded as the same continuous distribution. The same results when two FDCs of two different catchments built for the same time window are analysed. Thus, it is not possible to infer a FDC using parameters retrieved from the distribution of another FDC without considering the weather." The first sentence supports the assertion in the third, but the second does not. If two different catchments experience possibly similar weather but produce a different streamflow, the cause is not the weather.*

AC: We thank the reviewer for the comment. The sentences is now: "We show that two FDCs built for the same catchment with data corresponding to two different time windows, cannot be regarded as the same continuous distribution. This means that the FDCs cannot be considered an invariant characteristic of a basin. As other conditions did not substantially change across time, such as the land use, the reason should be the weather."

*R2: 11.b. Conclusions: p. 21, lines 13-14: "Since precipitation data series are characterized by a high number of zeros, here we used the Antecedent Precipitation Index (API)." This statement misses the more important fact that the API combines in a streamflow-like way the history of the precipitation. (A similar statement is made at the beginning of section 3.2 near the bottom of p. 8.)*

AC: We thank the reviewer for the suggestion. We have rephrased as:

"Since precipitation data series are characterized by a high number of zeros, here we used the Antecedent Precipitation Index (API). The API is used as it represents in a streamflow-like way the precipitation of the basin. It represents the memory of a basin as it provides the amount of precipitation released by the soil throughout the time."

*R2: 11.c. p. 22, lines 26-27: Qualitative statement about similarity in shape from beginning of Section 4 is repeated. This assertion needs to be quantified somehow.*

AC: A quantitative assessment of the goodness of the methodology is performed through the performance criteria. To describe why there can be a difference between the two FDCs, we rephrased the sentence as:

"The difference between the interpolated and observed FDCs can be due to the different temperature values characterizing the donor and target catchments."

[revised manuscript text omitted]
 a "destination" site through the monthly FDC of the destination site itself. The monthly FDC at the "destination" site is found using different methods such as (i) regionalization of FDCs based on available observed records from several adjacent gauges (Smakhtin et al., 1997) or (ii) conversion of FDCs calculated from monthly data into 1-day FDCs (Smakhtin, 1999). The procedure presented by Smakhtin and Masse (2000) is an extension of a previous work proposed by Hughes and Smakhtin (1996) to extend and/or filling in daily flow time series. The drawback of the procedures proposed by Hughes and Smakhtin (1996) and Smakhtin and Masse (2000) is the necessity of retrieving the monthly FDC of the target site with well-known literature methods before applying the methodology to extend the hydrograph. On the other hand, the novelty of the approach we propose is the possibility to retrieve the FDCs at partially ungauged sites from weather recorded at a donor catchment. Thus, without the need of applying procedures such as the regionalization. Our Aanalysis show that

[revised manuscript text omitted]

---

## Author Response (AR2)

The authors have presented a methodology to construct FDCs for missing portions of the hydrologic record at ungaged sites. The topic is very interesting and relevant. The authors have deeply considered all comments from the reviewers. To me, there is still a significant amount of work that needs to be completed before this manuscript can be published. There is nothing factually incorrect about the method, but I believe that it over-complicates itself and misses a few major elements. In particular, the derivation I have provided shows how this method is almost identical to previous publications. The revision should seek to explore the novelty of what is presented here.

*We acknowledge Dr. Farmer for the time he dedicated to our manuscript and we are thankful for his suggestions and comments that helped us to improve our paper. The main changes made in the manuscript based on Referee's comments:*

1. *We formulated the methodology and the underlying assumptions in a clearer manner in the Methodology section.*
2. *The German case study was removed to focus on the U.S.A. one.*

*In the following, we are reporting a detailed point-by-point reply to Dr. Farmer's comments. Authors reply to Reviewer's comments are in red Italic.*

As you can tell from the length of my review, I found this manuscript particularly engaging. (Sorry!) The main contribution of this work, in my opinion, is the hypothesis that the API can be used to produce the temporal sequence of non-exceedance probabilities of streamflow at a partially gaged site. In a related way, the authors hypothesize the existence of a unique, functional relationship between API and Q. This argues that the FDC is a function of weather, not just a function of the basin. These three points are phenomenally interesting to me, but the manuscript seems to get bogged down in the overly complex methodology rather than these hypotheses, which, when demonstrated, could be used to develop simplified methods. I would focus my revision on the novelty. I see the novelty as: (1) The FDC is a function of weather and location, not just location. (2) There exists a unique, one-to-one function relating API and streamflow. (3) For a given period, the temporal sequence of non-exceedance probabilities of streamflow is identical to the temporal sequence of non-exceedance probabilities of API. [NOTE: This is (I) on page 10.] (4) The temporal sequence of non-exceedance probabilities of API is identical at all sites in a region. [Note: This is (III) on page 10.] [NOTE FURTHER: Novelties 3 and 4 combine for an interesting result: QPPQ, described below.]

My main concerns are included below and pertain to the presentation and analysis of the methods. While these is nothing factually incorrect, the underpinnings and the implications of the method need to be explored and analyzed more fully.

To discuss the method, I'll begin by rephrasing the assumptions on page 10 in probabilistic terms, as they are central to this method:

I. $P(Q_{XY} < Q_{i,XY}) = P(API_{XY} < API_{i,XY})$, where X and Y designate the basin and time period, respectively. X and Y can therefore take a value of either D (donor) or T (target). The i indicates an arbitrary day of the record at the designated location and in the designated period.

II. $P(API_{DY} < API_{i,DY}) = P(Q_{TY} < Q_{i,TY})$ where all variables have been previously defined.

III. $P(API_{DY} < API_{i,DY}) = P(API_{TY} < API_{i,TY})$ where all variables have been defined previously.

$P(API_{DY} < API_{i,DY}) = P(API_{TY} < API_{i}, TY)$.

As an aside: It should be noted that (II) is an implication of (I) and (III). That is, if (I) and (III) are true, then II must be true. To be explicit: $P(Q_{TT} < Q_{i,TT}) = P(API_{TT} < API_{i,TT})$ by (I), and $P(API_{TT} < API_{i,TT}) = P(API_{DT} < API_{i,DT})$ by (III), so $P(Q_{TT} < Q_{i,TT}) = P(API_{DT} < API_{i,DT})$. This last equality is (III).

The method proposed by the authors relies on (I) and (III) and the assumption that there exists a one-to-one function $H\_X$ at each site that maps API to Q.

The method proposed by the authors is as follows: (The method is accurate, though I'm repeating it to put us in the same language.)

Given, $P(Q_{TT} < X) = 10.11\%$, find X.

By (I), $P(Q_{TT} < X) = P(API_{TT} < Y)$.

By (III), $P(API_{TT} < Y) = P(API_{DT} < Z)$.

So we know that $10.11\% = P(Q_{TT} < X) = P(API_{TT} < Y) = P(API_{DT} < Z)$, and Y and Z are knowable by observation. The method presented here ignores Y and focuses on Z.

From the appendix example, $Z = 37.72$ mm.

Because $H\_D(…)$ is a unique, one-to-one function that ingests API and transforms it to Q we can say that we are curious to know what Q is produced by the API Z at the donor site.

The method answers this question by looking at the donor basin in the donor period. That is, by (III), $P(API_{DD} < Z) = P(API_{TD} < Y)$. (Note, because $H\_D(…)$ and $H\_T(…)$ are one-to-one, unique, time-invariant functions there exists a constant mapping from Y to Z based on (III).)

By (I) we can then say further that $P(API_{TD} < Y) = P(Q_{TD} < X)$. Now, because we are in the donor period, X is knowable by observation given that we know $P(API_{DD} < Z)$.

From the appendix example, $P(API_{DD} < 37.72mm) = [7.52\%, 7.54\%]$.

The value X is therefore knowable by observation such that, $P(Q_{TD} < X) = [7.52\%, 7.54\%]$.

Because $H\_D(…)$ and $H\_T(…)$ are one-to-one, unique, time-invariant functions, Y-->X regardless of time period. So X is the value we were seeking. By observation, the value of X is 3.21mm.

There is nothing factually inaccurate about this method, but it does over-complicate the problem at a partially gaged site. If the target site (T) is partially gauged, then the assumptions provided above make the use of a donor unnecessary. This would proceed as:

Given, $P(Q_{TT} < X) = 10.11\%$, find X.

By (I), $P(Q_{TT} < X) = P(API_{TT} < Y)$.

Because all APIs at the target site in the target period, we can observe the value of Y.

We can then go to the donor period at the target basin with a known Y.

Because we assumed there existed a one-to-one, unique, time-invariant function $H\_T(…)$, and all the APIs and Qs of the donor period at the target basin are known, we can interpolate the value Y along $H\_T(…)$ as observed in the donor period to determine X. So there is no need for the donor basin when the target is partially (and sufficiently) gaged.

The need for a donor basin will be essential when you move to a completely ungaged basin.

However, with the ungaged basin, it is impossible to apply this method. This method relies on the donor period for an approximation of the FDC, though embodied through the function $H\_T(…)$, the proposed transferability of API across sites, and the proposed equality of probabilities for API and Q.

The reason that this method won't work at the fully ungaged site is because it relies on the donor period at the target site to estimate the FDC at the target site and uses the assumed $H\_T(…)$ to translate that across time periods (from donor to target).

To apply this method at the ungaged site, we will need an alternative method to approximate $H\_T(…)$. Arriving at this conclusion, it is obvious that this method, when applied to ungaged basins, is no different than the method presented by Smakhtin and his team in the mid to late

1990s. I will show this now, abbreviating their method as QPPQ:

Let's begin as we did before:

Given, $P(Q_{TT} < X) = 10.11\%$, find X.

By (I), $P(Q_{TT} < X) = P(API_{TT} < Y)$.

By (III), $P(API_{TT} < Y) = P(API_{DT} < Z)$.

Whereas earlier we stopped here, we can use (I) again to say that $P(API_{DT} < Z) = P(Q_{DT} < W)$.

We can then summarize this into a new implication of (I) and (III), namely $P(Q_{TT} < X) = P(Q_{DT} < W)$.

More generally, $P(Q_{DY} < Q_{i,DY}) = P(Q_{TY} < Q_{i,TY})$.

Now, if we had a first-order approximation of the relationship between Q and P at the target site, we could estimate X as a day in a simulated hydrograph.

The method presented by the present authors, uses API and $H_T(\dots)$ in the donor site to make this approximation. Smakhtin et al., and following work, use something like a regional-regression FDC. The QPPQ school uses this first-order FDC to produce a sequence of daily flows based on the probabilistic implication described here ($P(Q_{DY} < Q_{i,DY}) = P(Q_{TY} < Q_{i,TY})$). The QPPQ stops here.

You could move further to take the simulated hydrograph and compute a new FDC.

Because of inaccuracies used to produce the first-order FDC at the target site, inaccuracies stemming from estimation error, for example, and the evolving sequence of probabilities in the target period (for example, it could be overly weighted toward high flows). There is no guarantee that the first-order FDC used in QPPQ will be the same as the FDC that would be built from the resulting simulated hydrograph.

Moving to this second-order FDC would produce a time-dependent FDC at the ungauged site, much in the spirit of the methods presented by these authors.

These two processes, that of simplifying the method presented for partially ungaged sites and the extension of the QPPQ method, show that the work presented by these authors misses the novelty of their own method. I believe that the work of these authors has merit and the potential to advance the field. With further consideration of the implications and theoretical underpinnings of the proposed methodology, this manuscript could prove impactful. I will follow now with a few less-major concerns I have with the manuscript.

At the opening, it was necessary for me to rephrase the assumptions outlined on page 10. This is because those assumptions are not clear, and the tests provided to demonstrate their veracity test something different. In (I) it is assumed that the CDFs "correlate", and the test shows a temporal correlation between nonexceedance probabilities. While this isn't incorrect, it is ambiguous. For example, one could equally compute the correlation of API and Q quantiles to say that the CDFs correlate, but that would be very different than what the authors are seeking to assume or prove. Similarly, (III) states that the CDFs of API are identical across sites, but the proof shows only that the exceedance probabilities correlate in time. Identical CDFs would have the same probabilities associated with the same values of the proxy. These assumptions need to be less ambiguous.

As notes above, these assumptions are also redundant. Really, only (I) and (III) are needed, as (II) is a natural extension. Another natural extension, as shown above, is $P(Q_{DY} < Q_{i,DY}) = P(Q_{TY} < Q_{i,TY})$. This last one raises a point of confusion in the manuscript.

*We thank the reviewer for his interest in our paper. We have substantially modified the Methodology section to provide a clearer explanation of the approach and of the underlying assumptions. In the following, between quotes, we report the description that is also in the manuscript.*

"Formally the basic hypotheses of this paper are:

- Flow duration curves are not invariant properties of basins but are the product of basin, weather and human interactions. In this investigation we do not consider the human interactions.
- Precipitation is the most important influencing factor on discharge.
- Basins delay the reaction on precipitation therefore the API is a better indicator for the influence of precipitation on discharge.
- We assume that discharge and API are changing in a similar way for longer time periods.

Let $F_{A,T_i}(q)$ be the distribution of daily discharges at basin A and time period $T_i$ (flow duration curve for the selected time period) and $G_{A,T_i}(a)$ be the distribution of daily API at basin A and time period $T_i$.

The transformation from $T_i$ to $T_j$ provides an estimated $F^*_{A,T_j}(q)$:

$$F^*_{A,T_j}(q) = G_{A,T_j}(G^{-1}_{A,T_i}(F_{A,T_i}(q))) \tag{7}$$

This is a quantile/quantile transformation.

$$F_{A,T_j}(q) = F_{A,T_j}(F^{-1}_{A,T_i}(F_{A,T_i}(q))) \tag{8}$$

The basic question can be written in the form of the following equation

$$G_{A,T_j}(G^{-1}_{A,T_i}(p)) \approx F_{A,T_j}(F^{-1}_{A,T_i}(p)) \tag{9}$$

That can be summarized in the following question: do the percentiles of the API change in the same way as those of the discharge? Note that if the relationship between API and discharge is a good one then the two sides are nearly equal. Even a weak relationship can do a good job if the errors are independent and the sign of the change is correct. Figures 4 and 5 show the difference between the real change in percentiles and that obtained by using the API for different time periods according Eq.9. Note that the assumption that the FDC is time invariant would imply that the lines for the discharge are on the diagonal.

The correlation between API and discharge is around 0.6, but the transformations are quite similar and the API based transformation delivers good FDCs.

[Figure]

**Figure 4.** The transformation functions of the 1951-1960 FDC (solid) and API (dashed) to the target period 1971-1980 (blue) and 1981-1990 (orange) for Amite. For the sake of comparison, the diagonal is dotted in orange.

[Figure]

**Figure 5.** The transformation functions of the 1951-1960 FDC (solid) and API (dashed) to the target period 1971-1980 (blue) and 1981-1990 (orange) for Bogue. For the sake of comparison, the diagonal is dotted in orange.

If API is changing continuously in space then one can use the change of the FDC of a different location B for the estimation:

$$F_{A,T_j}^{**}(q) = F_{B,T_j}(F_{B,T_i}^{-1}(F_{A,T_i}(q))) \tag{10}"$$
* * *
On page 13, line 34, it is stated that "the FDC of a donor site cannot be transferred to another site". First, it is unclear what is mean by "the FDC": Do you mean the FDC in its entirety, including probabilities and quantiles, or the correlation as "proven" for (I), (II) and (III). In any case, the extension that $P(Q_{DY} < Q_{i,DY}) = P(Q_{TY} < Q_{i,TY})$ contradicts this statement. Furthermore, the authors then use this extension to transfer the FDC of a donor to a target in the German catchments. This confusion should be clarified. (In point of fact, I recommend removing the German catchments altogether.)

*We acknowledge the reviewer for his comment. We deleted the sentence as it was leading to confusion. We also removed the German case study for the sake of simplicity as suggested.*

A minor aside, the authors have no clearly defined that a partially gaged basin is. Is it any site with at least one value of streamflow? Is there some threshold for sufficiency? The method, whether as presented by the author or simplified here, requires a substantial donor period at the target catchment, but it might be important to define some length of record that would be required.

*A site is defined partially gauged when a limited amount of flow data is available. As we want to assess the performance of the method depending on the extension of the period with missing data, we have chosen time windows with different length as target periods and we have shown the performance of the method depending on this length. The minimum tested length is one-year long as it was used to retrieve the annual FDC. We clarified it in the Conclusion section.*

Stepping away from the methodology for a moment, I still struggled with the structure of the manuscript. Here, I have two recommendations. First, I suggest removing the section on energy and water limited catchments (2.3). We know that there are different limiters, and we know this is important, but I don't think this section adds to the manuscript. This is evidenced by the fact that the information in this section is never mentioned again in the manuscript: Not to explain the results or adapt the methodology. In a similar vein, I suggest removing the presentation and analysis of the German catchments. The results for the German catchments, starting on page 18, provide little value. The methodology is substantially different (confusingly using a streamflow proxy, as described above). The authors provide little discussion as to what new information is presented by these catchments. Obviously, with some additional discussion, both sections could prove useful, but it may be simpler just to remove them from the manuscript.

*We thank the reviewer for the comments, we deleted the sections related to the German catchments to keep the focus on the US catchments. On the other hand, we decided to keep the section on energy and water limited catchments as it involves a discussion on the role of the weather on annual runoff variability which is the leading message of the paper. We have added a discussion about energy and water limited catchments in the Conclusions section.*

While this revision did a great job of supporting its assumptions, I feel that a basic one was left out. The assumption that these exists the function $H_T(…)$ (page 9, line 7). As shown above, this is the crux of the methodology. I think it would be worthwhile to show that the interpolation of values implies that this relationship (between Q and API) is site dependent and time independent through a cross-validation exercise.

Finally, before considering minor comments, I was quite surprised that the authors did not address the large literature on the appropriateness of donor catchment selection. Instead, the examples presented here use the same donor for all gages and treat all gages as donors. Could it be that the method is sensitive to donor selection? In truth, the partially gaged method is probably not sensitive to donor selection, as demonstrated by the simplification where a donor is not needed, but the ungaged approach (QPPQ) is certainly dependent on donor selection.

*We thank the reviewer for the comment, the most important part is that both API and discharge change into the same direction, and the errors of the relationship are more or less independent so that they cancel out over time. The assumptions are now formulated in a clearer manner in the equations.*

MINOR COMMENTS:

Page 1, line 10: It seems odd to say that FDCs are "set up". They are a product of the record, they aren't established. Maybe something like, "The FDC of streamflow at a specific site provides knowledge on the distribution and characteristics of streamflow at that site."
*We rephrased as suggested.*

Page 1, line 12: "In spite of its importance,…"
*We rephrased as suggested.*

Page 1, line 14: What do you mean by partially gaged? The ability to build an FDC depends on this definition. Certainly, a site that has a record from 1980-2000 and 2002-2018 is partially gauged, but it still has more than enough information to build an FDC. I see your point that it is weather dependent, but it still leaves the definition of partially gaged ambiguous.
*The idea of this paper is that it is possible to retrieve the FDC of a basin using the precipitation; the unknown FDC could be annual or could be built with a longer data set. A site is defined partially gauged here if there is a lack of records in a given amount of time, this time could be a year or more. In the paper we show how the method performs choosing time windows with different length as target periods. Please, see also comment above regarding "partially gauged site" definition.*

Page 1, line 14: "…among the other streamgages."
*We were referring to other methods, it is clearer now.*

Page 2, line 3: I'm not sure that FDCs provide information on the severity because they don't talk about duration (length) of drought. So two sites could have a similar frequency of low flows, but one might see all low-flows consecutively (long drought), while the other might see intermittent lows (episodic drought). One could argue that drought is more severe in the former case.
*We removed the sentence.*

Page 2, line 12: "…to meet intake requests…"
Page 2, line 15, "As the FDC…"
*We rephrased as suggested.*

Page 2, line 27: "In spite of its importance, the FDC…"
*We rephrased as suggested.*

Page 3, line 27, "In the following…"
*We removed the sentence.*

Page 3, line 28, "…divided into water…"
*We removed the sentence.*

Page 3, line 28: Are you making a meaningful distinction between basins and catchments? If so, explain. If not, please use consistent terminology throughout.
*We checked the consistency and we are using basin now.*

Page 3, line 40: Check citation style.
Page 3, line 40: Revise to "The soil type can strongly affect the impact of climate on the water balance."
Page 3, line 41: I think the terms are karst and non-karst and should be throughout.

Page 4, line 2: "…karst regions makes it difficult to transfer information from ..."
Page 4, line 13: Please provide citation for degree-day approach.
*As we removed the German case study, we removed these sentences.*

Page 4, line 15: Keep in mind for formatting: This is an odd page break for a header.
*It is a mistaken format indeed.*

Page 5, line 21: "…the catchment allows for infiltration and stores…"
Page 5, line 21: "…runoff production declines since…"
Page 5, line 23: "However, the climatic timing…"
Page 5, line 24: "They show that the difference…"
*We rephrased as suggested.*

Page 8, line 30: I think maybe an extra sentence or two of explanation is needed here. Why would you expect two different time periods at the same site to be independent (as assumed by the KS test)? I see the point about auto-correlation, but I find it hard to imagine that the distribution of Fall 2012 streamflow and Fall 2013 stream flows are wholly independent at the same site. I think I follow your logic, and I would love to see it spelled out more fully. It may be tied to ambiguous definition of partial gaging.

*The KS test is designed to compare two distribution functions. The question is not whether the sample corresponding to different time periods are independent but whether they correspond to a different distribution. Anyhow the long memory is relatively low, and we consider full years thus annual cycles do not have an influence on our results. We clarified it in the text.*

Page 8, line 38: "…Figure 4 shows the magnitude of the difference between…"
Page 9, line 10: "…hydrological modelling as modelling often introduces additional errors…"
Page 9, line 11: "…errors and may be…"
*We rephrased as suggested.*

Page 9, line 15: Ask, defined above, denotes location, what location is being addressed here?
 *We rephrased to introduce the location.*

Page 10, line 10: This paragraph cites discussion about why the alpha should decay with time. Does it not decay here? It's fixed at 0.85?

*We have deleted the sentence that was leading to confusion. In this paragraph we describe that alpha may assume values ranging from 0 to 1, depending on the type of API's memory one wants to capture. Once that the value of alpha is chosen, this is constant throughout the time. Here it is set equal to 0.85.*

Page 10, line 17: See discussion above. Here you need to clarify that you are talking about the correlation between $p(Q<Q_i)$ and $P(API < API_i)$ for the day i. This is different than the correlation of $Q_i$ and $API_i$ of rank i.
Page 10, line 27: Are these Pearson correlations? Please specify?
*We consistently revised the methodology section.*

Page 13, line 5: Here you are using "," to denote a decimal point and "." to denote figures to the right of the decimal. This is not the style of the journal. I suggest, "X.yz, where y and z are the digits to the right to the decimal" Perhaps still better: In binning by percentiles, all percentages were rounded down to the nearest whole number.
*We rephrased as suggested.*

Page 13, line 9: Shouldn't there be an absolute value for this statement to be true? Bias is defined here as an average. If, for example, we looked at three days producing these values of the thing in parenthesis: -3, 0, 3. In this case, the BIAS would be 0, but this is certainly not a perfect fit.
*We have rephrased the description of BIAS.*

Page 13, line 13: "…and so is the MAE. It …"
 *We rephrased as suggested.*

Page 13, line 29: I think it would be better to show the p-value of D*. Looking at Figure 8 (page 14, line 12), the D may be bigger, but it may not be as significant because of changing sample size.

*There was a typo regarding the D* values that are now correct. The p-value is always zero but for a few cases that are reported in the captions and in the text.*

Page 17, line 13: I find it very confusing that the color scale changes from figure to figure (e.g. 9 and 10). The change makes it hard to compare across figures.

*We have corrected the scale so that it is easier to compare now, we also carefully checked the figures since there was an issue regarding the colors of Figure 10 (now Figure 11). Now it is fixed.*

Page 18, line 6: Here the proxy is donor streamflow. I discuss this above, but this strikes me as odd when page 13, line 34 seems to say you should not do this. (As you know, I think it might be best to just remove this section.)
*We removed it as suggested.*

Page 19, line 2: The correlation between the donor and the target is problematic and cannot be known a priori (line 12). This needs to be contextualized in an ungaged example that is not discussed here. The ambiguity here makes the conclusion on line 39 of page 40 rather tenuous.

*We removed the German case study as suggested and so this paragraph, besides the methodology is now presented in a clearer manner.*

Page 23, line 18: The probability statement should be in terms of API, right?
*Yes, it is. Thank you, it was a typo.*

[revised manuscript text omitted]
 c̶a̶t̶c̶h̶m̶e̶n̶t̶basin is Blanco (TX). Each target c̶a̶t̶c̶h̶m̶e̶n̶t̶basin is indicated a̶b̶o̶v̶e̶ in the corresponding box. Negative values of the NSE as well as outliers of BIAS and MAE are reported o̶n̶ in the corresponding box.

When both target and donor periods equal 15 years, the agreement between interpolated and observed flow values is high, Figure 1̶0̶11. The NSE shows values of efficiency around 1, thus there is a good match between interpolated and observed values, even though there are few exceptions. The errors are very low in value, as shown by the MAE, which also reveals a poor performance for high flows, while the performance improves for intermediate and low flows. The high flows are more likely o̶v̶e̶r̶e̶s̶t̶i̶m̶a̶t̶e̶d̶estimated with a higher error, w̶h̶i̶l̶e̶ than intermediate and low flows a̶r̶e̶ ̶m̶o̶r̶e̶ ̶l̶i̶k̶e̶l̶y̶ u̶n̶d̶e̶r̶e̶s̶t̶i̶m̶a̶t̶e̶d̶ as also shown by the BIAS.

[revised manuscript text omitted]